# Stony coral tissue loss disease induces transcriptional signatures of in situ degradation of dysfunctional Symbiodiniaceae

Kelsey M. Beavers [1], Emily W. Van Buren[1], Ashley M. Rossin[2], Madison A. Emery[1], Alex J. Veglia [3], Carly E. Karrick [3], Nicholas J. MacKnight [1], Bradford A. Dimos[1], Sonora S. Meiling[4], Tyler B. Smith [4], Amy Apprill [5], Erinn M. Muller [6], Daniel M. Holstein [2], Adrienne M. S. Correa [3], Marilyn E. Brandt [4] & Laura D. Mydlarz[1] ✉

Stony coral tissue loss disease (SCTLD), one of the most pervasive and virulent coral diseases on record, affects over 22 species of reef-building coral and is decimating reefs throughout the Caribbean. To understand how different coral species and their algal symbionts (family Symbiodiniaceae) respond to this disease, we examine the gene expression profiles of colonies of five species of coral from a SCTLD transmission experiment. The included species vary in their purported susceptibilities to SCTLD, and we use this to inform gene expression analyses of both the coral animal and their Symbiodiniaceae. We identify orthologous coral genes exhibiting lineage-specific differences in expression that correlate to disease susceptibility, as well as genes that are differentially expressed in all coral species in response to SCTLD infection. We find that SCTLD infection induces increased expression of *rab7*, an established marker of in situ degradation of dysfunctional Symbiodiniaceae, in all coral species accompanied by genus-level shifts in Symbiodiniaceae photosystem and metabolism gene expression. Overall, our results indicate that SCTLD infection induces symbiophagy across coral species and that the severity of disease is influenced by Symbiodiniaceae identity.

Global climate change has initiated a dramatic increase in the prevalence, frequency, and severity of marine disease outbreaks[1,2], and has contributed to whole reef ecosystem regime shifts as well as significant population decreases of certain coral species[3]. As a hotspot for coral diseases, the Caribbean is particularly at risk of severe biodiversity loss, exhibiting a 50–80% decline in living coral tissue since the 1970s[4,5]. Although the etiologic agents of many of these diseases remain elusive, one prevailing hypothesis postulates that a growing number of diseases result from environmentally induced microbiome imbalances and a subsequent increase in opportunistic or polymicrobial infections[6,7]. As climate change pressures continue to escalate, emerging and endemic outbreaks in the Caribbean are

[1]Biology Department, University of Texas at Arlington, Arlington, TX, USA. [2]Department of Oceanography and Coastal Sciences, Louisiana State University, Baton Rouge, LA, USA. [3]Department of BioSciences, Rice University, Houston, TX, USA. [4]Center for Marine and Environmental Studies, University of the Virgin Islands, St. Thomas, USVI, USA. [5]Marine Chemistry and Geochemistry Department, Woods Hole Oceanographic Institution, Woods Hole, MA, USA. [6]Mote Marine Laboratory, Sarasota, FL, USA. ✉e-mail: mydlarz@uta.edu

likely to decimate reef-building coral populations, resulting in reef ecosystem collapse.

Stony coral colonies are holobionts comprised of the coral host and a diverse consortium of symbiotic microorganisms, including Symbiodiniaceae, bacteria, fungi, and viruses[8,9]. The coral host provides a protective habitat as well as metabolic byproducts to its symbiotic partners and in return receives oxygen and organic compounds from its Symbiodiniaceae, as well as pathogen defense and nutrient cycling from its other microbiota[8,9]. Symbiodiniaceae and bacteria symbionts are important to colony health, but anthropogenic disturbance can destabilize these associations, initiate bleaching (of Symbiodiniaceae) and ultimately result in colony tissue loss[3,10].

The Caribbean is currently experiencing an outbreak of the most pervasive and contagious coral disease on record—stony coral tissue loss disease (SCTLD)[11]. Originating off the coast of southeast Florida in 2014, SCTLD is a waterborne disease known to cause acute tissue loss in more than half of reef-building species in the Caribbean[12,13]. Although previous work on SCTLD suggests that there may be common secondary bacterial infections[14–17], and antibiotic treatment has been shown to be 84% effective in stopping tissue loss[18], a pathogen responsible for SCTLD has not yet been identified. However, a growing line of evidence implicates viral infection of Symbiodiniaceae in the etiology of this disease based on (1) a reduction or halting of lesion progression in bleached corals[19], (2) histopathological examination identifying lytic necrosis of host gastrodermal cells where Symbiodiniaceae reside[20], (3) transmission electron microscopy (TEM) detection of filamentous viral-like particles associated with endosymbiont pathology[21], and (4) the assembly of putative filamentous viral genomes from SCTLD-affected coral holobiont metatranscriptomes[22].

SCTLD affects over 22 species of coral and previous evidence suggests that coral holobionts vary in their susceptibilities to this disease[12,23]. In April of 2019, three months after initial cases of SCTLD were observed in the United States Virgin Islands (USVI), Meiling et al.[23] conducted a controlled SCTLD transmission experiment in which fragments from six species of coral were split in half and placed into control and treatment mesocosms. Colony halves placed into treatment mesocosms were incubated with SCTLD-infected colonies of *Diploria labyrinthiformis*, while corresponding genotype fragments placed into control mesocosms were incubated with visually healthy colonies of *D. labyrinthiformis*. By monitoring lesion appearance over an 8-day experimental period, Meiling et al.[23] was able to obtain tangible disease phenotype measurements, such as disease prevalence and incidence, relative risk of lesion development, and lesion growth rate. These phenotypes were used to demonstrate microbial community shifts across different coral microhabitats following disease exposure and lesion appearance[24]. Overall, results presented in Meiling et al.[23] found that of the six species tested, *Colpophyllia natans* and *Orbicella annularis* showed the greatest susceptibility to SCTLD and the highest lesion growth rates in the USVI compared to *Pseudodiploria strigosa*, *Porites astreoides*, and *Montastraea cavernosa*.

Here, we utilize the spectrum of disease phenotypes obtained by Meiling et al.[23] and apply comparative transcriptomics to elucidate the biological processes underlying variations in SCTLD susceptibility. Transcriptomics provides an invaluable tool to measure gene expression responses to pathological conditions, and a previous study has linked SCTLD progression in two coral species to the expression of genes involved in immunity, apoptosis, and tissue rearrangement[25]. However, the cellular mechanisms by which different coral species and symbiont lineages (presented here at the genus level) respond to this disease remain largely unknown. By linking the gene expression profiles of multiple coral species to quantitative disease phenotypes, we identify coral and Symbiodiniaceae expression shifts in response to SCTLD infection that contribute to variation in disease susceptibility and resistance.

## Results

### Transmission experiment

At the end of the transmission experiment, 100% of the *C. natans* and *O. annularis* fragments showed signs of active SCTLD lesions, followed by 75% of *P. strigosa* fragments, 62.5% of *P. astreoides* fragments, and 37.5% of *M. cavernosa* fragments. Coral fragments were classified by their treatment outcome as either "controls", "exposed" or "infected." Fragments exposed to apparently healthy *D. labyrinthiformis* donor corals were classified as controls. Fragments exposed to SCTLD-infected *D. labyrinthiformis* donor corals but did not develop lesions were classified as exposed, while those that did develop lesions were classified as infected. *C. natans* had the highest median relative risk of infection (12.19), followed by *O. annularis* (12.09), *P. strigosa* (9.30), *P. astreoides* (7.91) and *M. cavernosa* (5.10) (Fig. 1a). *C. natans* also had the fastest average lesion growth rate (0.42 cm²/h), followed by *O. annularis* (0.20 cm²/h), *P. strigosa* (0.13 cm²/h), *M. cavernosa* (0.09 cm²/h), and *P. astreoides* (0.03 cm²/h) (Fig. 1a). Full experimental results are presented in Meiling et al.[23] and can be found in Supplementary Data 1 and 2.

### Coral transcriptome assembly and annotation

To determine differential expression, we sequenced 52 transmission experiment coral tissue samples which resulted in a total of 2.82 billion raw reads with an average of 54.3 million reads per sample. Reference de novo metatranscriptomes for *C. natans* and *P. astreoides* as well as a genome-guided *M. cavernosa* assembly were sourced from previous Mydlarz lab work[26]. Genome-guided transcriptome assembly of the *O. annularis* cleaned reads produced an assembly of 34,741 contigs with a N50 size of 7601 bp and de novo transcriptome assembly of the *P. strigosa* cleaned, quality-filtered, coral-only reads produced an assembly of 23,116 contigs with a N50 size of 4970 bp (Table 1).

### Isolation and quantification of coral and Symbiodiniaceae reads

From the clean reads, a total of 1.02 billion mapped to the coral host and 887.6 million reads mapped to the sample's dominant symbiont lineage (see Methods—Isolation and quantification of coral and Symbiodiniaceae reads), with an average of 19.58 million and 17.07 million reads per sample mapped to the host and dominant symbiont, respectively (Supplementary Data 1). Mapping of coral host and dominant symbiont reads to their respective transcriptome and subsequent transcript quantification resulted in a total of 13,692 *C. natans*, 13,138 *O. annularis*, 9796 *P. strigosa*, 13,271 *P. astreoides*, 11,628 *M. cavernosa*, 10,675 *Symbiodinium* spp., 10,072 *Breviolum* spp., 11,939 *Cladocopium* spp., and 10,328 *Durusdinium* spp. length-normalized transcripts with an annotation evalue of $1.0e^{-6}$. Dominant Symbiodiniaceae identified for each sample based on transcripts largely agreed with Symbiodiniaceae lineages identified from each sample via Illumina MiSeq amplicon sequencing of the internal transcribed spacer-2 (ITS-2) region of Symbiodiniaceae rDNA (Supplementary Information —Fig. 2, Supplementary Data 3).

### Differential expression of coral transcripts

The number of genes significantly expressed (padj ≤ 0.05) in response to SCTLD exposure varied among species. The most susceptible species, *C. natans*, had the highest number of differentially expressed genes (DEGs) between exposed and non-exposed treatments at 1350 followed by *M. cavernosa*, the least susceptible species, at 385 DEGs, *O. annularis* at 229 DEGs, *P. strigosa* at 90 DEGs and *P. astreoides* at 42 DEGs (Fig. 1b). We identified the highest percentage of DEGs involved in immunity and/or the response to viral infection in the highly susceptible species *O. annularis* (7% of DEGs), followed by *C. natans* and *P. astreoides* (5% of DEGs), *M. cavernosa* (4% of DEGs) and *P. strigosa* (2% of DEGs) (Fig. 1b). No DEGs were shared across all 5 species.

Merging of Uniprot Entry IDs across coral species followed by DESeq2 normalization and low average expression removal resulted in

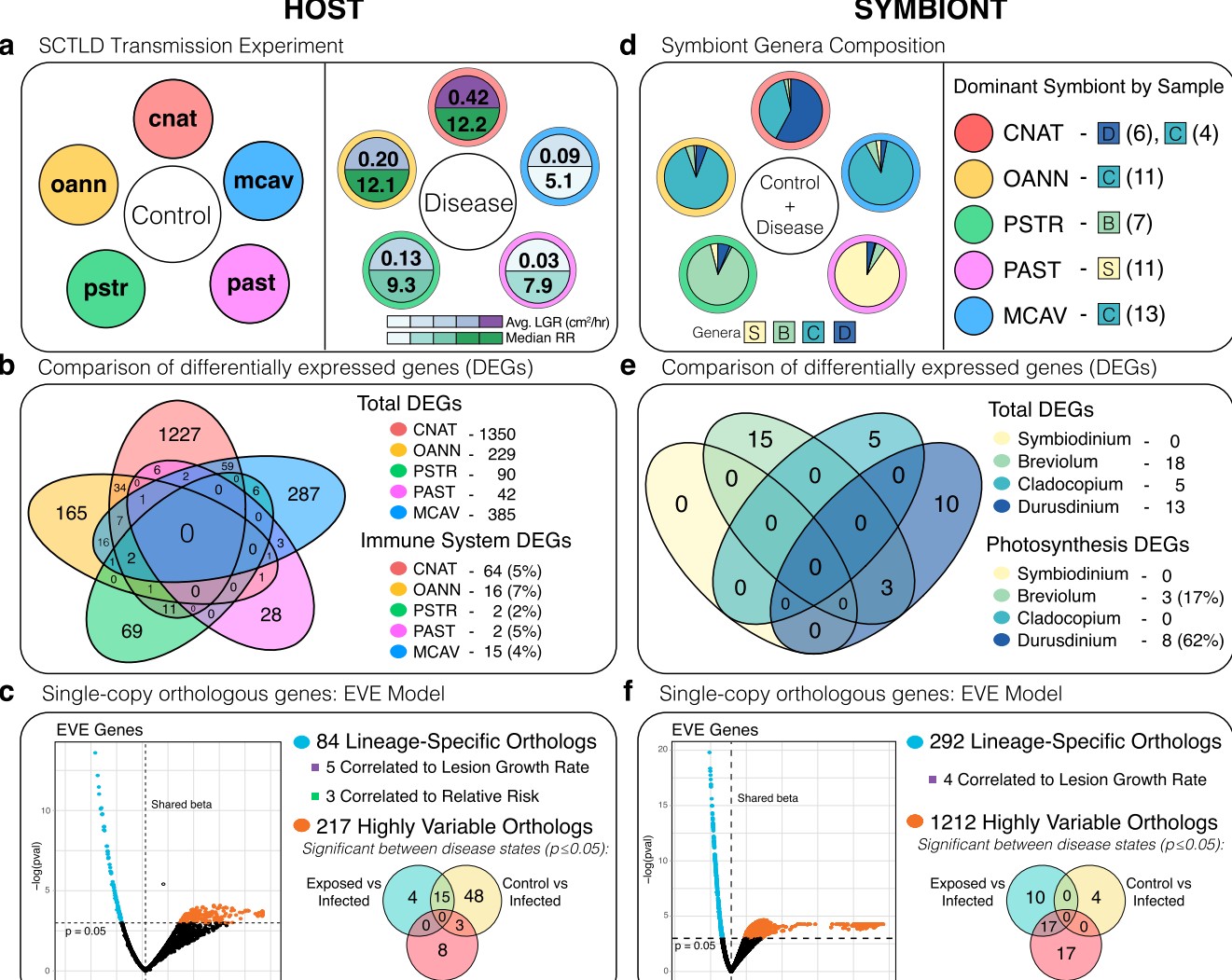

**Fig. 1 | Overview of Experimental Design and Analysis. a** Summary of the SCTLD transmission experiment in which eight replicates of five susceptible coral species were split in half, with one half exposed to a healthy *D. labyrinthiformis* colony (control) and the other half exposed to a diseased *D. labyrinthiformis* colony (disease). The control panel names the species in the study, and the disease panel shows the resulting species-level disease phenotypes used to inform our statistical analyses. **b** Venn Diagram showing the number of unique and shared DEGs across species between control and disease treatments. The total number of DEGs within each species was enumerated as well as the number and percentage of those involved in immunity and viral response. **c** EVE model summary for host orthologs. 1766 single-copy orthologs were identified across the five species in our study and their expression was used as inputs for the EVE model used to differentiate genes with lineage-specific and highly variable expression. Pearson correlations were run on lineage-specific orthologs to identify those correlated to disease phenotypes, and one-way ANOVAs followed by TukeyHSD tests were run on highly variable orthologs to identify those with significant differential expression between disease states. **d** Symbiont composition determined by binning RNAseq reads to

*Symbiodinium* (S), *Breviolum* (B), *Cladocopium* (C) and *Durusdinium* (D) reference transcriptomes. The first panel shows symbiont genera composition pie charts averaged across samples within each species. The second panel shows the dominant symbiont present within each sample. The expression from these dominant symbionts were used for downstream analyses. **e** Venn Diagram showing the number of unique and shared DEGs across dominant symbionts between control and disease treatments. The total number of DEGs within each was enumerated as well as the number and percentage of those involved in the relevant process of photosynthesis. **f** EVE model summary for dominant symbiont orthologs. 5125 single-copy orthologs were identified across symbiont genera and their expression was used as input for the EVE model. Pearson correlations were run on lineage-specific orthologs to identify those correlated to disease phenotypes, and one-way ANOVAs followed by TukeyHSD tests were run on highly variable orthologs to identify those with significant differential expression between disease states. (cnat *Colphophyllia natans*, oann *Orbicella annularis*, pstr *Pseudodiploria strigosa*, past *Porites astreoides*, mcav *Montastraea cavernosa*, LGR Lesion Growth Rate, RR Relative Risk, ECM Extracellular Matrix, EVE Expression Variance and Evolution).

2147 inferred homologs with measurable expression. Of these, 150 were identified by GO term as involved in immunity and/or the response to viral infection and 17 were identified as involved in extracellular matrix (ECM) structure. The one-way ANOVA identified 16 immune/viral response homologs and 3 ECM homologs with significant differential expression between disease states (TukeyHSD Test; $p \leq 0.05$) (Supplementary Data 7). Among these homologs included the antiviral gene Interferon regulator factor 2 (*IRF2*), two genes putatively involved in symbiont population maintenance, Ras-

related protein rab 7 (*rab7*) and Polyunsaturated fatty acid 5-lipoxygenase (*ALOX5*), and a collagen alpha chain fragment (*COA*).

Merging of Orthogroup IDs across coral species followed by DESeq2 normalization and low average expression removal resulted in 4759 annotated orthologs with measurable expression. Of these, 212 were identified by GO term as involved in immunity and/or the response to viral infection and 107 were identified as involved in extracellular matrix structure. The one-way ANOVA identified 30 immune/viral response orthologs and 18 ECM orthologs with

**Table 1 | Reference transcriptome sources and assembly metrics**

| Species | Type | No. contigs | Complete & single copy | Complete and duplicated | Fragmented | Missing | N50 | Percent annotated |
|---|---|---|---|---|---|---|---|---|
| *Coral host transcriptome assembly metrics based on metazoan reference* | | | | | | | | |
| *C. natans* | de novo | 50,582 | 631 | 86 | 115 | 146 | 12,255 | 27% |
| *M. cavernosa* | genome-guided[82] | 38,865 | 709 | 75 | 91 | 103 | 8027 | 30% |
| *O. annularis* | genome-guided[56] | 34,741 | 719 | 99 | 95 | 65 | 7601 | 38% |
| *P. astreoides* | de novo | 37,167 | 806 | 44 | 60 | 68 | 6457 | 36% |
| *P. strigosa* | de novo | 23,116 | 603 | 104 | 79 | 192 | 4970 | 42% |
| *Symbiodiniaceae transcriptome assembly metrics based on eukaryote reference* | | | | | | | | |
| *S. CassKB8* | de novo | 72,152 | 123 | 6 | 39 | 87 | 17,845 | 15% |
| *B. minutum* | de novo | 51,199 | 172 | 6 | 19 | 58 | 11,053 | 20% |
| *C. goreaui* | de novo | 65,838 | 110 | 66 | 18 | 61 | 18,032 | 18% |
| *D. trenchii* | de novo | 82,273 | 173 | 13 | 15 | 54 | 17,347 | 13% |

Reference transcriptomes for *M. cavernosa*, *C. natans*, and *P. astreoides* were sourced from Dimos et al.[26]. Reference transcriptomes for *S. CassKB8*, *B. minutum*, *C. goreaui* and *D. trenchii* were sourced from Bayer et al.[83], Parkinson et al.[84], Davies et al.[85], and Bellantuono et al.[86], respectively. The "Percent annotated" column shows the percentage of protein-coding transcripts that were annotated with an evalue cutoff of $1.0e^{-6}$.

significant differential expression between disease states (TukeyHSD Test; $p \le 0.05$) (Supplementary Data 9). Three of these homologs, SMAD family member 6 (*smad6*), Toll-like receptor 6 (*TLR6*), and TNF-receptor-associated factor 3 (*Traf3*), were identified as members of the nuclear factor kappa B (NF-κB) pathway (Fig. 2). Deleted in malignant brain tumors 1 (*Dmbt1*), a gene involved in mucosal innate immunity, was significantly downregulated in infected corals relative to controls ($p = 0.0019$) (Fig. 2). *Rab7* was also identified by our ortholog analysis as significantly upregulated in infected corals relative to controls ($p = 0.023$), as well as Charged multivesicular body protein 4b (*Chmp4b*), another gene involved in the same endosome maturation pathway as *rab7* ($p = 0.0017$) (Fig. 2). Among the ECM orthologs significantly differentially expressed between control and infected corals included Superoxide dismutase [Cu-Zn] 1 (*sodA*) and Alpha-2 type I collagen (*col2a1*) (Fig. 2).

**Coral expression variance and evolution model**

The differential expression of single-copy orthologs across the five coral species were used to test variation both among and within species by applying the Expression Variance and Evolution (EVE) model[27]. A total of 1817 single-copy orthologs were identified between the five coral species, 1766 of which had measurable expression levels across all species and were used as inputs for the EVE model. This model, a phylogenetic ANOVA, parametrizes the ratio (β) of population to evolutionary expression level. A large β is associated with higher variation within versus between species and indicates expression diversity or plasticity. A small β is associated with higher variation between versus within species and indicates lineage-specific expression diversity. B does not change if there is stabilizing or no selection acting on expression level. Here we use EVE to identify two categories of single-copy orthologous genes: (1) those with small β values ($p \le 0.05$) exhibiting lineage-specific or adaptive expression and (2) those with large β values ($p \le 0.05$) exhibiting expression plasticity. EVE identified 84 significant lineage-specific orthologs and 217 significant highly variable orthologs in the coral animal ($p \le 0.05$) (Fig. 1c).

Of the 84 lineage-specific orthologs, 75 were annotated with a sufficient evalue ($1.0e^{-6}$) and kept for analysis. The expression of those 75 orthologs were averaged across species and tested for linear correlation with: (1) species median relative risk of infection and (2) species average lesion growth rate using Pearson's correlation. Three orthologs were identified as significantly correlated to relative risk ($p \le 0.05$) and were annotated as Vacuolar protein-sorting-associated protein 35 (*vps25*), Zeta-sarcoglycan (*SGCZ*), and Cytoplasmic dynein 1 light intermediate chain 2 (*DYNC1LI2*) (Fig. 3). Additionally, five orthologs were identified as significantly correlated to lesion growth rate ($p \le 0.05$) and were annotated as Arrestin domain-containing protein 2 (*ARRDC2*), COP9 signalosome complex subunit 8 (*Cops8*), Golgin subfamily A member 7B (*GOLGA7B*), Ribulose-phosphate 3-epimerase (*Rpe*), and PHD finger protein 10 (*phf10*) (Fig. 3). *ARRDC2*, located in the cytoplasmic vesicle and plasma membrane and predicted to be involved in protein transport[28], showed a highly significant positive linear correlation to lesion growth rate ($R = 0.99$, $p = 0.0011$) (Fig. 3). Another gene, *Cops8*, a member of the COP9 signalosome, showed significant negative correlation between expression level and lesion growth rate ($R = -0.91$, $p = 0.032$) (Fig. 3).

Of the 217 highly variable orthologs, the one-way ANOVA identified 78 with significant differential expression between disease states ($p \le 0.05$), of which 53 were annotated with a sufficient evalue ($1.0e^{-6}$). The relative expression of those annotated orthologs with the most significant differences in expression ($p \le 0.01$) were averaged across disease state and plotted in a heatmap (Fig. 4a). Notable orthologs with highly significant differences in expression between control and SCTLD-infected corals were plotted by species as well as disease state (Fig. 4b). Of all the highly variable orthologs, Tropomyosin alpha-4 chain (*Tpm4*) showed the most significant difference in expression between control and infected corals ($p = 2.0e-07$). Additionally, Transmembrane prolyl 4-hydroxylase (*P4HTM*), an enzyme that "senses" hypoxic conditions, was also significantly downregulated in corals infected with SCTLD (Fig. 4). Alternatively, two orthologs significantly upregulated between control and SCTLD-infected corals were annotated as Protein BTG1 (*BTG1*) and Parkin coregulated gene protein homolog (*PACRG*) (Fig. 4).

***C. natans* differential expression**

PCA of transcriptome-wide gene expression of *C. natans* samples identified dominant symbiont genus as the main driver (PC1) of expression-level differences in this species rather than disease state (Supplementary Information—Fig. 4A). Because *C. natans* samples dominated by *Durusdinium* symbionts experienced greater disease severity as measured by lesion growth rate compared with *Cladocopium*, the top 20 genes driving the differences in expression between *Cladocopium*- and *Durusdinium*-dominated hosts (PC1 loadings) were identified and their rlog expression was plotted (Supplementary Information—Fig. 4B).

**Differential expression of Symbiodiniaceae transcripts**

The number of genes significantly differentially expressed (padj ≤ 0.05) in response to SCTLD exposure varied between dominant

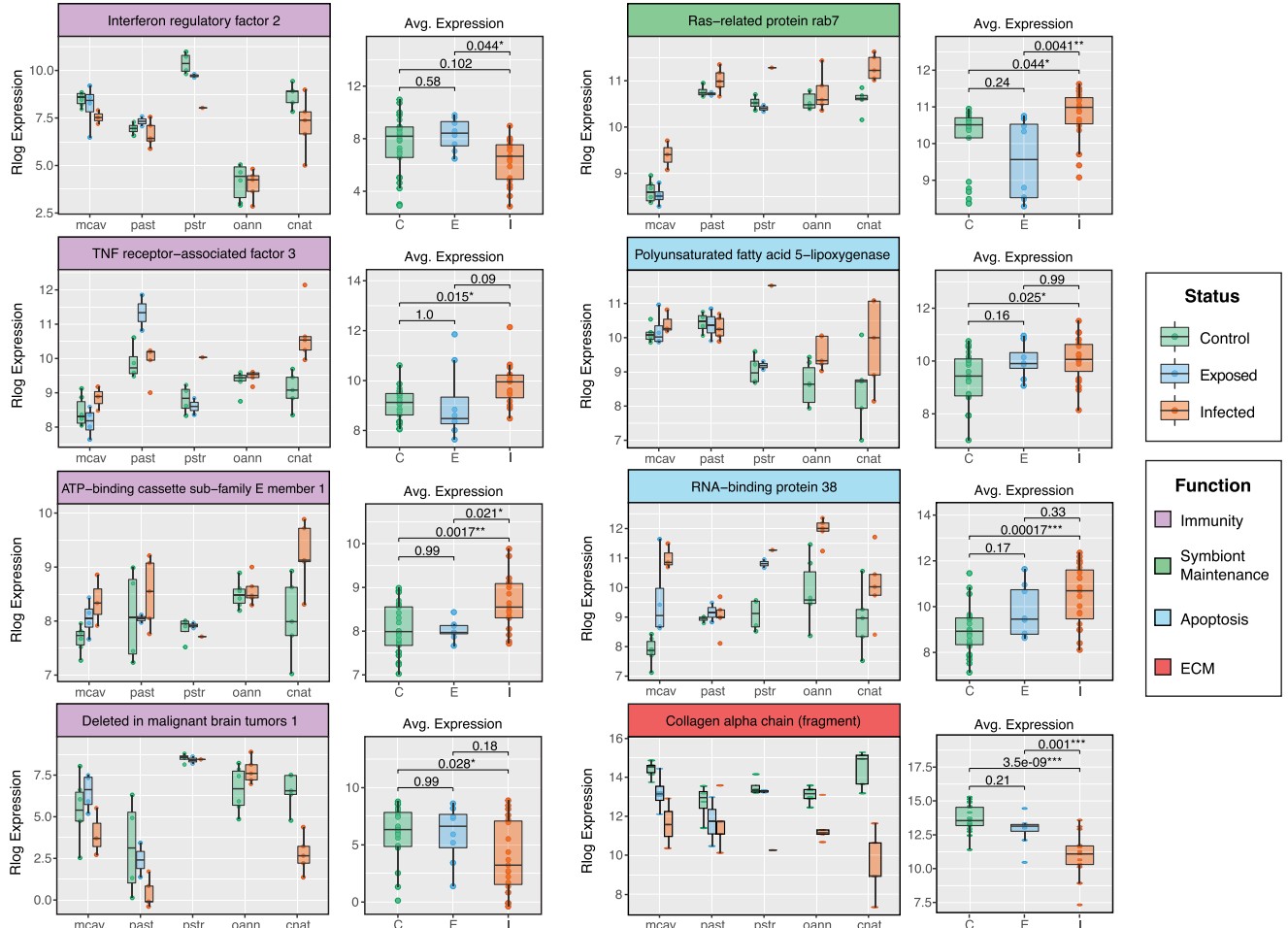

**Fig. 2 | Expression Boxplots of Relevant Orthologs with Significant Differential Expression Between Disease States.** Eight of the orthologs identified by our immune and extracellular matrix (ECM) GO term search that exhibit significant differential expression between disease states (TukeyHSD; $p \le 0.05$). Boxplots show the rlog transformed expression of relevant orthologs in each sample, organized by coral species on the left and by disease status on the right. Color of gene headers correspond to biological function obtained from literature searches and GO terms. Color of boxplots correspond to experimental disease status: C Control ($n = 25$), E Exposed (no lesion, $n = 8$), and I Infected (with lesion, $n = 19$). P-values represent TukeyHSD results following one-sided ANOVA tests (*) = $p \le 0.05$; (**) = $p \le 0.01$, (***) = $p \le 0.001$. Boxplot elements: center line, median; box limits, upper and lower quartiles; whiskers, 1.5x interquartile range; points beyond whiskers, outliers). Source data are provided as a Source Data file.

symbiont genera. *Breviolum* spp. had the highest number of DEGs with 18, followed by *Durusdinium* spp. with 13 DEGs, and *Cladocopium* with 5 DEGs. *Symbiodinium* spp. had no DEGs (Fig. 1). No DEGs were shared amongst all three of the genera that had DEGs, but *Breviolum* spp. and *Durusdinium* spp. shared three: Photosystem I P700 chlorophyll a apoprotein A1 (*psaA*), Photosystem II CP43 reaction center protein (*psbC*), and Photosystem II CP47 reaction center protein (*psbB*) (Fig. 5). To enumerate the number of DEGs in each genus involved in photosynthesis, DEGs containing the GO term "photosynthesis" were counted. Remarkably, 62% of *Durusdinium* spp. DEGs were involved in photosynthesis, followed by 17% of *Breviolum* spp. DEGs and 0% of *Cladocopium* spp. DEGs (Fig. 1e).

**Symbiodiniaceae expression variance and evolution model**
The differential expression of single-copy orthologs across the four dominant symbiont genera were used to test variation both among and within species by applying the EVE model[27]. A total of 5332 single-copy orthologs were identified between the four dominant symbiont genera, 5125 of which had measurable expression levels across all genera and were used as inputs for the EVE model. EVE identified 292 significant lineage-specific orthologs and 1212 significant highly variable orthologs ($p \le 0.05$) (Fig. 1). Of the 292 lineage-specific orthologs,

120 were annotated with a sufficient evalue ($1.0e^{-6}$) and kept for analysis. Four of those annotated orthologs were identified as significantly correlated to lesion growth rate ($p \le 0.01$): Probable protein phosphatase 2C 45 ($R = -0.993$, $p = 0.008$), Charged multivesicular body protein 3 ($R = -0.991$, $p = 0.009$), Zinc finger CCCH domain-containing protein 1 ($R = -0.999$, $p = 0.001$), and Serine/threonine protein phosphatase 2A regulatory subunit β ($R = -0.996$, $p = 0.004$).

Of the 1212 highly variable orthologs, the one-way ANOVA identified 48 with significant differential expression between disease states ($p \le 0.05$), of which 24 were annotated with a sufficient evalue ($1.0e^{-6}$). 75% (18/24) of these orthologs exhibit a significant transcriptional shift in the SCTLD-exposed (but no lesion) corals relative to controls, while only 4.2% (1/24) exhibit such a shift in the SCTLD-infected (with lesion) corals (Fig. 6a). Relevant orthologs were plotted by species as well as disease state (Fig. 6b). Two orthologs, annotated as Cyclin-dependent kinase 2 (*Cdk2*), a cell cycle regulator that orchestrates the entry into mitosis/meiosis[28], and 2-alkenal reductase (*DBR*), involved in the degradation of reactive carbonyl species[28], showed significant downregulation in the SCTLD-exposed corals relative to controls (Fig. 6b). Alternatively, Guanine Deaminase (*GuaD*), a gene involved in the production of xanthine and ammonia from guanine, exhibited a significant increase in expression in the SCTLD-exposed corals relative to controls (Fig. 6b).

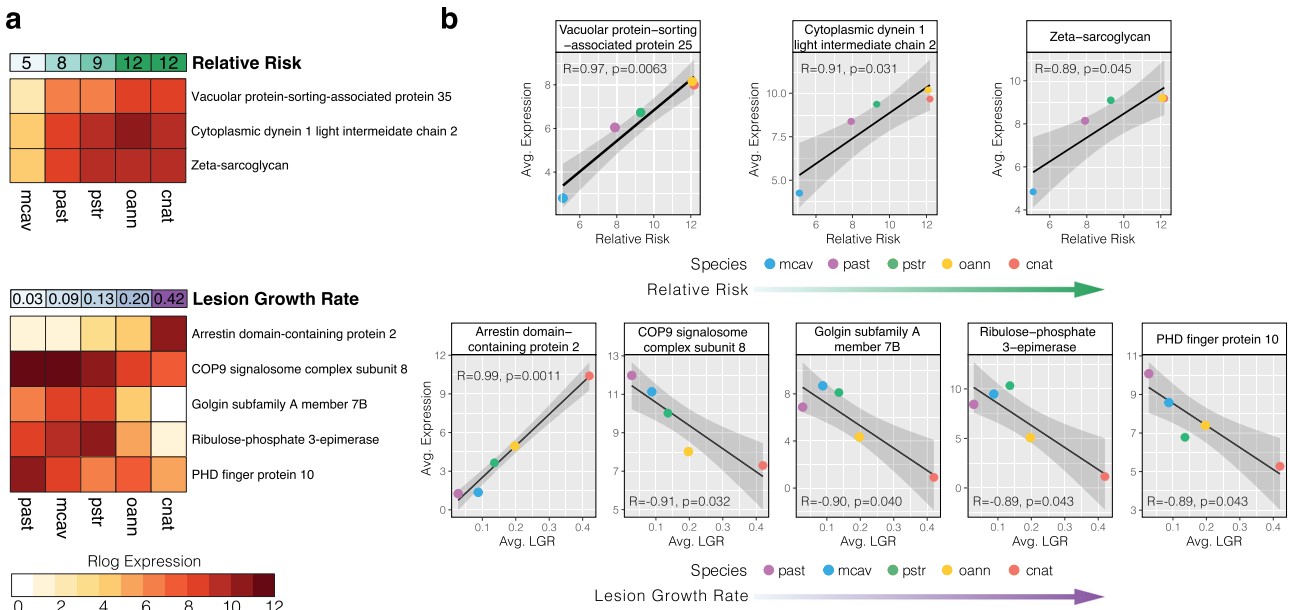

**Fig. 3 | Expression of single-copy coral orthologs significantly correlated to disease phenotypes.** The eight single-copy orthologs identified by EVE as lineage-specific (exhibiting low expression plasticity within species, but high variation between species) with significant correlation to species-level disease phenotypes (Pearson correlation; $p \leq 0.05$, $R < -0.85$ or $R > 0.85$). **a** Heatmaps plot the rlog transformed expression levels of each ortholog averaged across coral species. **b** Scatterplots plot this expression on the y-axis and the disease phenotype on the x-axis. One-sided Pearson correlation coefficients and P-values are provided for the linear trendline (black), and the gray area represents a 95% confidence interval.

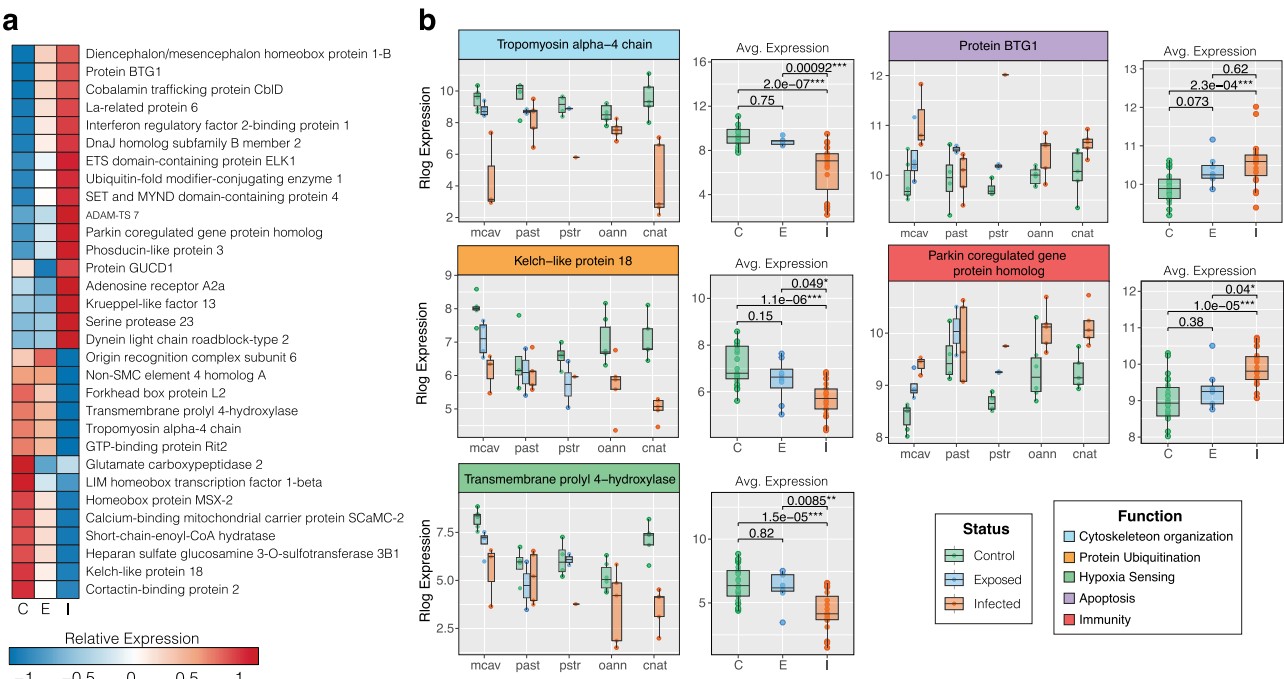

**Fig. 4 | Single-copy coral orthologs with significant differences in expression between disease states.** The 31 annotated single-copy orthologs identified by EVE as highly variable (exhibiting expression plasticity across species and samples) that show the most significant differential expression between disease states (TukeyHSD; $p \leq 0.01$). **a** Heatmap plots the relative expression of those 31 highly variable orthologs averaged across disease state: C Control ($n = 25$), E Exposed (no lesion, $n = 8$), and I Infected (with lesion, $n = 19$). **b** Boxplots show the rlog transformed expression of relevant orthologs in each sample, organized by coral species on the left and by disease status on the right. Color of gene headers correspond to biological function obtained from literature searches and Uniprot. Color of boxplots correspond to experimental disease status. P-values represent TukeyHSD results following one-sided ANOVA tests (*) = $p \leq 0.05$; (**) = $p \leq 0.01$, (***) = $p \leq 0.001$. Boxplot elements: center line, median; box limits, upper and lower quartiles; whiskers, 1.5x interquartile range; points beyond whiskers, outliers).

## Coral *rab7* correlations to Symbiodiniaceae genes

Two photosystem genes in *Symbiodinium* (*psaA* and *psbC*) and in *Durusdinium* (*psaA* and *psbB*) were significantly positively correlated to coral *rab7* expression, an established marker of symbiophagy ($p \leq 0.05$) (Fig. 7). Although not statistically significant, the expression of Superoxide dismutase and Heat shock protein 70 were also positively correlated to coral *rab7* expression in these symbiont genera. Alternatively, in *Cladocopium* symbionts the expression of Heat shock

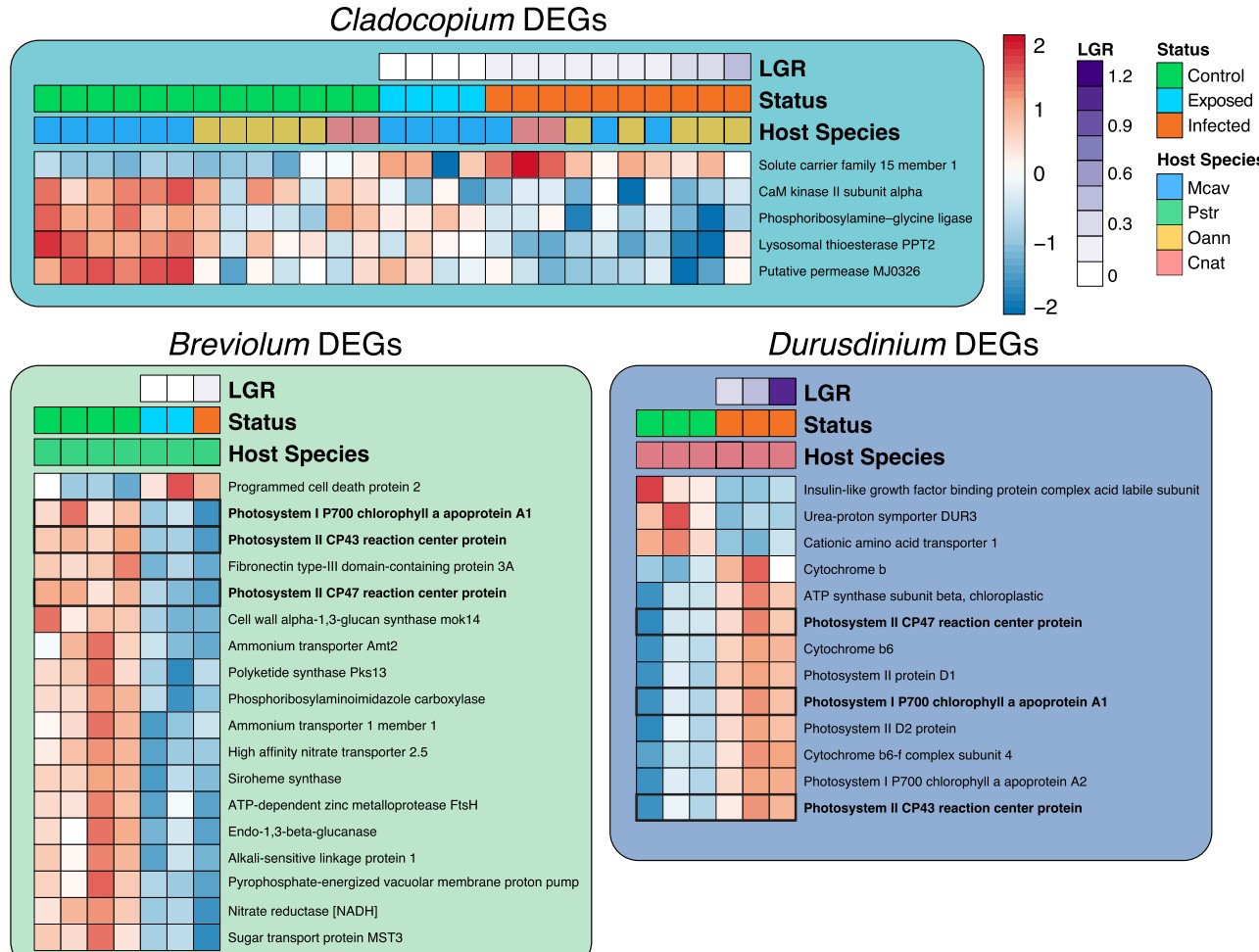

**Fig. 5 | Dominant Symbiont DEGs.** Heatmaps showing the relative expression of the significantly differentially expressed genes (control vs. disease treated) from each genus of dominant symbiont. Dominant symbionts belonging to the genera *Breviolum* shared three DEGs with dominant symbionts belonging to the genera *Durusdinium* (boxed rows and bolded gene names). No DEGs were identified in dominant symbionts belonging to the genera *Symbiodinium*. (LGR Lesion Growth Rate).

protein 70 was significantly negatively correlated to coral *rab7* expression ($p \leq 0.01$) (Fig. 7). No *Breviolum* genes were significantly correlated to coral *rab7* expression.

### Coral *rab7* correlations to histology measurements
*Rab7* expression was significantly negatively correlated to average symbiont size ($R = -0.48$, $p = 0.00049$), but was not correlated to the ratio of symbiont size to symbiosome size ($R = -0.19$, $p = 0.19$) (Fig. 8). There was no significant difference in average symbiont size between disease states, but there was a significant decrease in the ratio of symbiont size to symbiosome size in both the disease-exposed and disease-infected fragments relative to the controls ($p = 5.7e-04$ and $p = 3.83e-05$, respectively) (Fig. 8).

## Discussion
Caribbean coral reefs are currently experiencing the worst disease-related mortality event on record, with several reef-building species suffering massive regional die-offs. Here, we investigated the gene expression responses of five phylogenetically distinct coral species and their dominant Symbiodiniaceae following experimental exposure to SCTLD. Through the combined use of differential expression analyses on immune-related genes and highly variable orthologs, as well as correlation analyses of lineage-specific orthologs to disease phenotypes, we examined the gene expression patterns consistent across

species as well as those that may contribute to variations in species susceptibility. First, we found that SCTLD infection induced expression of homologous and orthologous genes involved in immunity, apoptosis, and ECM structure in all five coral species. Second, we showed that species-level SCTLD susceptibility was correlated to lineage-specific differences in expression of single-copy orthologs involved in vesicular trafficking and signal transduction. Third, we found evidence that SCTLD exposure (without visible lesions) induces major expression shifts in the highly variable Symbiodiniaceae single-copy orthologs, but not in those of the coral animal. Finally, we identified and compared the genes significantly differentially expressed by various Symbiodiniaceae genera in response to SCTLD infection and posit that SCTLD disrupts normal host-symbiont interactions and in situ degradation of dysfunctional Symbiodiniaceae.

### SCTLD infection induces an immune response across coral species
Consistent with multiple lines of evidence implicating SCTLD as a viral disease, we found transcriptional signatures of antiviral immunity across five coral species experimentally infected with SCTLD. Two members of the interferon antiviral response pathway, *Traf3* and *IRF2*, showed significant differential expression between disease states. *Traf3*, a negative regulator of the nuclear factor kappa B (NF-κB) pathway and a positive regulator of type I interferon (IFN1) production

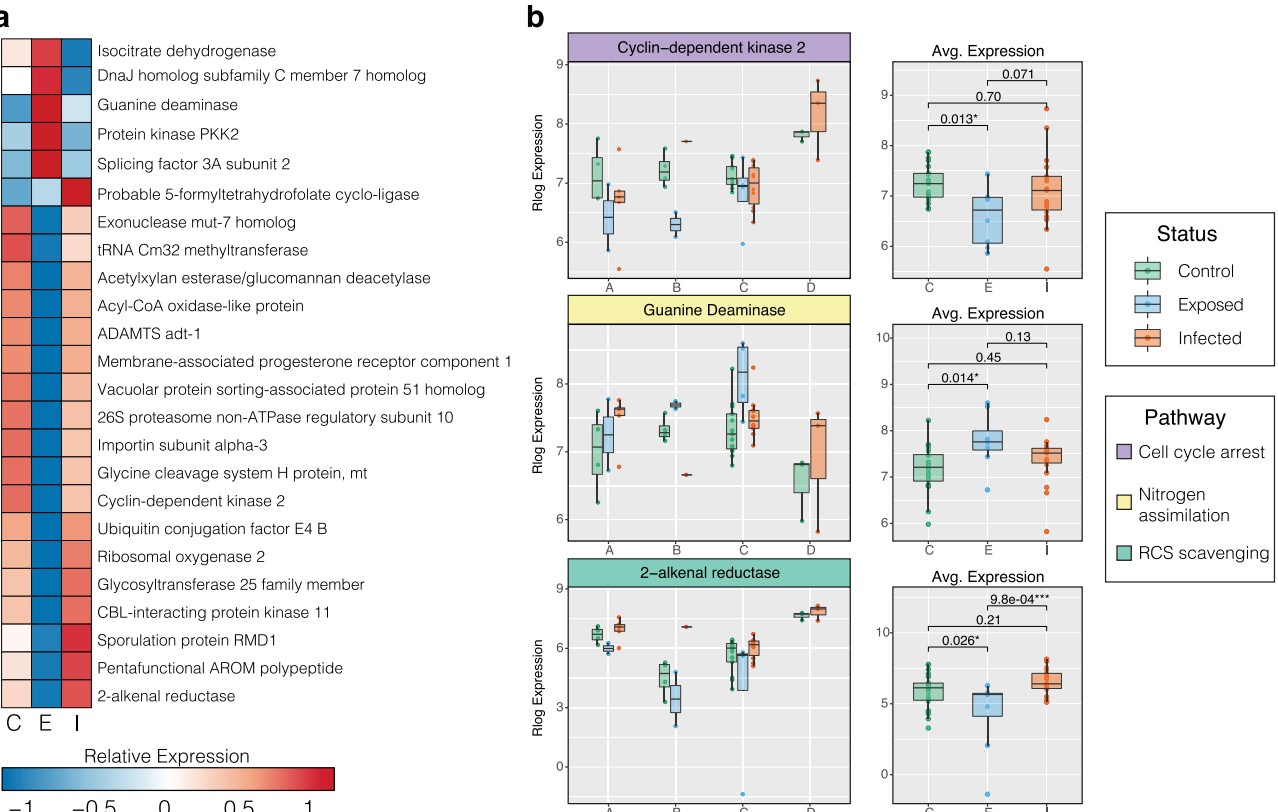

**Fig. 6 | Single-copy symbiont orthologs with significant differences in expression between disease states.** The 24 annotated single-copy orthologs identified by EVE as highly variable (exhibiting expression plasticity across genera and samples) that show significant differential expression between disease states (TukeyHSD; $p \leq 0.05$). **a** Heatmap plots the relative expression of those 26 highly variable orthologs averaged across disease state: C Control ($n = 24$), E Exposed (no lesion, $n = 8$), and I Infected (with lesion, $n = 19$). **b** Boxplots show the rlog transformed expression of relevant orthologs in each sample, organized by dominant symbiont genera on the left and by disease status on the right. Color of gene headers correspond to biological function obtained from literature searches and Uniprot. P-values represent TukeyHSD results following one-sided ANOVA tests (*) = $p \leq 0.05$; (**) = $p \leq 0.01$, (***) = $p \leq 0.001$. (A = *Symbiodinium*, B = *Breviolum*, C = *Cladocopium*, D = *Durusdinium*). Boxplot elements: center line, median; box limits, upper and lower quartiles; whiskers, 1.5x interquartile range; points beyond whiskers, outliers).

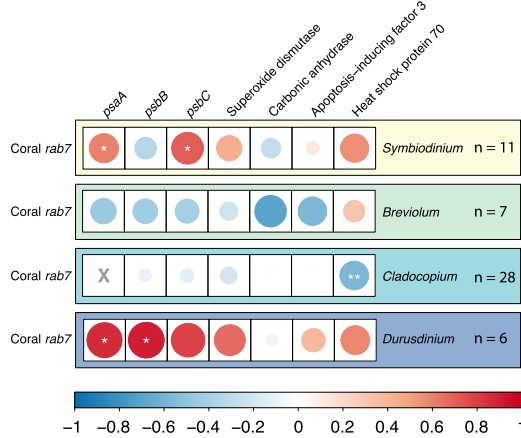

**Fig. 7 | *Rab7* Correlations.** Correlation matrix showing the pairwise correlation coefficients (R values) between coral *rab7* expression and the expression of nine relevant symbiont genes. The color of the circle corresponds to the *R* values in the scale and the size of the circle corresponds to the absolute value of that R value. Significant correlations are present with stars (*) = $p \leq 0.05$; (**) = $p \leq 0.01$, (***) = $p \leq 0.001$. Actual *p*-values can be found in Supplementary Data 14.

in humans[29], was significantly upregulated in SCTLD-infected corals relative to control. Similarly, *IRF2*, an antagonist of IFN1 transcriptional activation[28], was downregulated in SCTLD-infected corals relative to resistant corals. The upregulation of IFN1 production and the

concordant downregulation of its antagonist is a strong indication that corals manifesting visible SCTLD lesions are activating an antiviral immune response.

We also found transcriptional evidence of a functional shift in the coral surface mucus layer, the first line of defense against foreign particles and microbes[30]. *Dmbt1*, involved in mucosal innate immunity and microbial homeostasis in humans[28], exhibited significant overall downregulation in SCTLD-infected corals relative to controls. Wright et al.[31] found downregulation of this gene in *Acropora millepora* challenged with *Vibrio* spp., leading them to the hypothesis that this gene may play a role in maintaining healthy associations with commensal microbes. Our results are consistent with this hypothesis, as downregulation of *Dmbt1* in SCTLD-infected corals was accompanied by significant shifts in mucus microbiome composition towards dysbiosis in specimens from this same study[24]. Downregulation of *Dmbt1* in SCTLD-infected corals may therefore render corals unable to maintain mucosal microbial homeostasis, leading to a loss of the protective capabilities of their surface mucus layer and making them susceptible to secondary infection by opportunistic bacteria. These results explain why antibiotic treatment is sufficient to arrest treated SCTLD lesions, but ineffective at preventing new lesion appearance on other parts of the same coral colony[32].

In addition, our results indicate a significant upregulation of apoptosis and related stress-response processes in SCTLD-infected corals relative to controls. *SMAD6*, a member of the transforming growth factor β (TGF-β) signaling pathway, exhibited a strong decrease in expression between control and infected corals. Knockdown of this

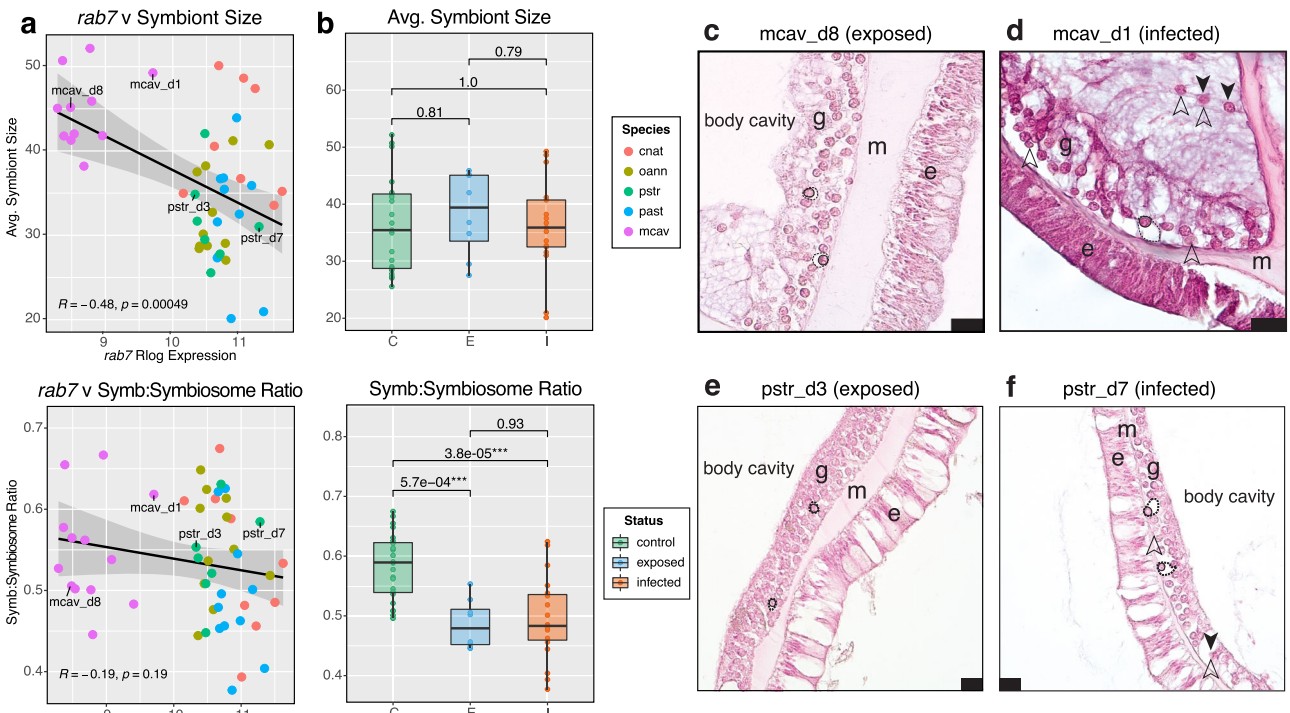

**Fig. 8 | Histology and *rab7* correlation analysis. a** Scatterplots show *rab7* rlog transformed expression on the *x*-axis plotted against the histology phenotype on the *y*-axis. Correlation coefficients and *P*-values are provided for the linear trendline (black), and the gray area represents a 95% confidence interval. **b** Boxplots show the histology phenotypes plotted by disease status: C Control (*n* = 25), E Exposed (no lesion, *n* = 8), and I Infected (with lesion, *n* = 19). *P*-values represent TukeyHSD results following one-sided ANOVA tests (*) = $p \leq 0.05$; (**) = $p \leq 0.01$, (***) = $p \leq 0.001$. **c** visually healthy *M. cavernosa* exposed to SCTLD. **d** SCTLD-infected *M.* *cavernosa* with elevated *rab7* expression **e** visually healthy *P. strigosa* exposed to SCTLD. **f** SCTLD-infected *P. strigosa* with elevated *rab7* expression. e epidermis, m mesoglea, g gastrodermis. Dashed ovals outline symbiosomes, solid ovals outline symbiont cells within symbiosomes; solid arrows = exocytosed symbiont cells; empty arrows = potentially degraded symbiont cells. Scale bar = 20 μm. Boxplot elements: center line, median; box limits, upper and lower quartiles; whiskers, 1.5x interquartile range; points beyond whiskers, outliers.

---

protein has been shown to increase apoptosis and inhibit cell cycle progression in human cells[33]. Similarly, *ALOX5*, a proinflammatory gene hypothesized to play a role in the phagocytosis of apoptotic bodies in regenerating *Hydra*[34], showed a concomitant increase in expression in infected corals. We also found significant upregulation of the anti-oxidant *sodA*, an established coral immune response antioxidant also involved in symbiosis breakdown[35,36]. Finally, we observed strong downregulation of multiple collagen genes in infected corals, suggesting a stress-induced decrease in ECM structural proteins not immediately needed for cell survival. These results are highly similar to the expression level shifts of immune, apoptosis, and ECM genes detected in SCTLD-infected corals from Florida[25].

### SCTLD susceptibility is correlated to increased vesicular trafficking and decreased signaling

Our results indicate that species-level variation in SCTLD susceptibility correlates to differential expression of single-copy orthologs involved in vesicular trafficking and signaling. Two orthologs involved in vesicular transport, *DYNC1LI2* and *vps25*, displayed a positive linear correlation between expression level and relative risk of SCTLD infection. *DYNC1LI2* acts as a motor for the retrograde transport of vesicles along microtubules and *vps25* is a component of the endosomal sorting complex required for transport II (ESCRT-II), which functions in the endocytosis of ubiquitinated membrane proteins for lysosomal degradation[28]. Interestingly, some RNA viruses such as HIV-1 hijack the ESCRT pathway to facilitate viral budding and egress from infected cells[37]. In fact, ESCRT-II depletion or elimination in human cells resulted in reduced HIV-1 Gag protein production, visible decreases in budding efficiency, and diminished intracellular levels of virion

protein[38]. The increased average expression of *vps25* in highly susceptible coral species may represent an elevated capacity of a SCTLD-associated virus to hijack ESCRT-II to facilitate virion production and export.

Additionally, our results indicate that signal transduction is reduced in coral species exhibiting the most severe forms of SCTLD lesion progression. The expression of one single-copy ortholog annotated as *ARRDC2* showed a strong positive correlation to lesion growth rate. Mammalian β-arrestins are known to regulate G-protein-coupled receptor (GPCR) trafficking and signaling, and proteins with predicted "arrestin-like" domains have been indicated to function similarly[39]. Interestingly, the human bacterial pathogen *Streptococcus pneumoniae*, which binds to a GPCR to enter host cells for transcytosis, can avoid lysosomal degradation in cells exhibiting overexpression of β-arrestin 1[40]. Bacterial work on SCTLD suggests that there may be common secondary bacterial infections[14–17], and antibiotic treatment has been shown to halt lesion progression[18,32]. The strong correlation between *ARRDC2* expression and lesion growth rate in SCTLD corals indicates that overexpression of this gene may prevent successful lysosomal degradation of bacteria, potentially opportunistic pathogens invading compromised tissue.

Another single-copy ortholog involved in signaling, annotated as *Cops8*, showed significant negative correlation between expression level and lesion growth rate. *Cops8* is a member of the COP9 signalosome, a conserved multiprotein complex that functions as an important regulator of many signaling pathways through its control over ubiquitin-proteasome-mediated protein degradation[41]. Through the ubiquitin-dependent degradation of the NF-κB inhibitor IkappaBalpha, the COP9 signalosome can mediate NF-κB activation,

which in turn triggers an immune response[42]. Therefore, these results indicate that species of coral exhibiting rapid lesion progression of SCTLD, such as *C. natans* and *O. annularis*, may be unable to mount an effective immune response due to their comparatively low levels of immune signaling.

## SCTLD-infected corals are reducing endosymbiont density through in situ degradation

One of the most significant findings of this study is the consistent differential expression of genes involved in symbiophagy, the in situ degradation of dead or dysfunctional symbionts. This process, triggered by a stress event, transforms the host-derived symbiosome from an arrested state of phagocytosis into a digestive organelle resulting in the consumption of the endosymbiont[43–46]. Protein rab7, a recognized marker of autophagy across taxa, is associated with endocytic phagosomes containing dead or dysfunctional zooxanthellae in the anemone *Aiptasia pulchella* and stony coral *Pocillopora damicornis*[43,44]. In our study, we found *rab7* to show significant upregulation in infected corals relative to both control and exposed (no lesion) corals, implicating symbiophagy as a common cellular phenotype of SCTLD lesions. Similarly, we observed strong increases in expression of *Chmp4b*, a gene involved in the same endosome-to-lysosome maturation pathway as *rab7*. Additionally, *ALOX5*, a pro-inflammatory gene whose expression was found by Fuess et al.[47] to be negatively correlated with symbiont density, was also significantly upregulated in infected corals relative to controls. The upregulation of *rab7, Chmp4b*, and *ALOX5* in SCTLD-infected corals corroborates strongly with evidence that this disease causes a breakdown of host-symbiont physiology[20].

Because in situ degradation of endosymbionts represents a host innate immune response to compromised symbionts[46], we investigated transcriptional evidence of symbiont impairment. Within the symbiont DEGs, we see evidence of photosynthesis dysfunction in both the downregulation of photosystem genes in *Breviolum* spp., as well as the upregulation of those same genes in *Durusdinium* spp. In fact, all the genes upregulated by *Durusdinium* symbionts are involved in energy production, likely indicative of photosystem overexcitation. Furthermore, the samples dominated by *Durusdinium* spp. displayed the fastest lesion growth rates in our study, indicating that symbiont photosystem overexcitation may lead to worse outcomes for SCTLD infected corals.

Coral *rab7* expression was positively correlated to symbiont photosystem genes in both *Symbiodinium* and *Durusdinium*, as well as the antioxidant gene Superoxide dismutase (*sodA*) and Heat shock 70 kDa protein (*Hsp70*)—two known indicators of symbiont stress and dysfunction[46,48]. Because coral *rab7* is an established marker of symbiophagy, these correlations indicate that symbiont photosystem overexcitation and stress are associated with higher levels of in situ degradation of symbionts by their coral host. This pattern, however, is not ubiquitous across symbiont genera and potentially contributes to variability in disease susceptibility. Using histology measurements, we found high *rab7* expression associated with small symbiont size, although there was no significant difference in symbiont size across disease states. However, there was a significant decrease in the ratio of symbiont size to symbiosome size in both the disease-exposed and disease-infected corals relative to controls. Taken together, these results indicate that SCTLD exposure triggers symbiont stress and subsequent symbiosome enlargement, a change in morphology indicative of symbiophagy[44].

Expression-level shifts in the coral single-copy orthologs identified by EVE as highly variable provide evidence as to why diseased corals may be inducing symbiophagy rather than exocytosis to remove dysfunctional symbionts. Of all highly variable orthologs, *Tpm4* showed the most significant difference in expression between control and infected corals. The protein encoded by this gene contributes to most, if not all, functions of the actin cytoskeleton[28], and has been shown by single-cell sequencing to be localized predominantly within alga-hosting cells in corals[49]. In fact, Mayfield et al.[50] found decreased expression of a related gene, *tropomyosin*, in the Pacific coral *Seriatopora hystrix* at night, supporting their hypothesis that coral alga-hosting cells experience major cytoskeletal rearrangement as endosymbionts switch from photosynthesis to respiration. Additionally, the strong upregulation in infected corals of *Btg1*, an antiproliferation protein maximal during the $G_0/G_1$ phase of the cell cycle[28], indicates that SCTLD infection induces host cell quiescence or senescence, stalling cell growth and proliferation. Cells exposed to environmental stress are thought to enter $G_0$ due to lack of nutrients necessary for division[51]. Because corals derive up to 100% of their daily metabolic requirements from their Symbiodiniaceae[52], it is likely that the concomitant expression decrease in *Tpm4* and increase in *Btg1* are indicative of coral starvation due to inadequate photosynthate transfer from symbionts. Taken together with the increased expression of coral immune- and stress-response genes, this provides evidence that SCTLD-infected corals are mounting an autophagic immune response against dysfunctional symbionts and/or their pathogens.

Contrary to the coral EVE results, the majority of the highly variable *Symbiodiniaceae* single-copy orthologs identified by EVE exhibit significant transcriptional changes in the SCTLD-exposed (but not lesioned) symbionts relative to controls. The dramatic shift in symbiont gene expression seen in the exposed corals could represent two things: (1) early symbiont responses to SCTLD infection before the manifestation of visible lesions on the coral, or (2) a successful symbiont response to SCTLD pathogen exposure preventing the onset of disease. Because the expression of many of these orthologs is comparable between control and SCTLD-infected corals, we believe the latter to be true. If so, the downregulation of the cell cycle regulator Cyclin-dependent kinase 2 (*Cdk2*) and upregulation of the ammonium assimilator guanine deaminase (*GuaD*) in SCTLD-exposed algal cells provides evidence that symbiont cell cycle arrest may prevent the onset of visible lesions in the host. This is consistent with results from Huntley et al.[24] showing that mucus microbiome alterations occurred even in the corals exposed to SCTLD but did not develop lesions. Up to half of the fixed carbon supplied by Symbiodiniaceae to the coral host is expelled as mucus[53,54], making it plausible that symbiont cell cycle arrest has metabolic consequences negatively affecting coral mucus production.

The results presented here provide evidence that (1) viral infection of Symbiodiniaceae is implicated in SCTLD pathology, and (2) disease manifestation is associated with in situ degradation of defective symbionts. Our study provides the underlying mechanisms involved in processes identified by histological observations, such as shrunken and necrosing symbionts, enlarged symbiosomes, and viral-like particles uniquely localized within algal endosymbiont cells[20,21,23]. Our application of innovative bioinformatic methods allows us to consider both coral and algal endosymbiont processes involved in SCTLD infection and brings us closer to understanding the pathogenicity of SCTLD so we may effectively combat this devastating disease.

## Methods

### Experimental approach, sample preparation and sequencing

All research conducted in this study complies with all relevant ethical regulations as outlined by the Department of Planning and Natural Resources Coastal Zone Management. The SCTLD transmission experiment was carried out at the University of the Virgin Islands (UVI) in April of 2019 and is published in Meiling et al.[23]. Briefly, one fragment from each of five species of stony coral (*C. natans, M. cavernosa, O. annularis, P. astreoides*, and *P. strigosa*) was placed into a control mesocosm equidistant from a central healthy *D. labyrinthiformis* donor coral colony. Corresponding genet fragments from each species

were placed into an experimental mesocosm equidistant from a SCTLD-infected *D. labyrinthiformis* donor coral colony. Eight genets of healthy *D. labyrinthiformis* were used as control donor corals and eight genets of diseased *D. labyrinthiformis* were used as SCTLD donor corals. This paired design was replicated for a total of eight genets per species. Experimental coral fragments that developed lesions were removed when 30% tissue loss was achieved and were stored at −80 °C for RNA sequencing. Corresponding control genet fragments were removed and processed at the same time. All fragments regardless of health status were processed at the end of an eight-day experimental period. Upon completion of the experiment, lesion growth rate was determined for each SCTLD-infected fragment and median relative risk of infection was calculated for each species by Meiling et al.[23] (Supplementary Data 2).

Total RNA was extracted using the RNAqueous-4PCR Total RNA Isolation Kit from Invitrogen (Life Technologies AM1914). First, a sterilized bone cutter was used to scrape off a pea-sized amount of frozen coral tissue from each fragment into a 2 mL microcentrifuge tube. On diseased fragments, tissue was harvested between 2 and 5 cm from the visible lesion. Tissue samples were lysed with a refrigerated Qiagen Tissuelyser II at 30 oscillations/s for 30 s during the lysis stage and the elution stage was performed in two steps (30ul elution followed by another 30 µl elution). Contaminating DNA and chromatin were removed from each sample of total RNA using the Ambion DNase I (RNase-free) kit from Invitrogen (Life Technologies AM2222). Prior to sequencing, all samples underwent quality assessment using an Agilent Bioanalyzer 2100 at the University of Texas at Arlington Genomics Core Facility (Arlington, TX, USA). The 52 samples that passed quality assessment (RIN numbers ≥ 7) were sent to Novogene Co., LTD (Beijing, China). At Novogene, samples were preprocessed for mRNA enrichment using polyA tail capture and mRNA libraries were prepared using the NEBNext Ultra II RNA Library Prep Kit from Illumina. The resulting cDNA libraries were fed into the Illumina NovaSeq 6000 for 150 bp, paired-end sequencing.

### Coral transcriptome assembly and annotation
Raw reads from Novogene were quality filtered using FastP v. 0.20.1[55] under default parameters. A genome-guided transcriptome for *O. annularis* was generated using the *Orbicella faveolata* genome[56] and a de novo metatranscriptome was assembled for *P. strigosa*, both using Trinity v. 2.11.0[57]. Coral-only transcripts were obtained from the de novo metatranscriptomes (*C. natans*, *P. astreoides* and *P. strigosa*) adopting the *in-silico* filtration method outlined by Davies et al[58]. First, the longest transcript isoform was obtained using the get_longest_isoform_seq_per_trinity_gene.pl script available within the Trinity v. 2.11.0 package[59]. Next, this assembly was blasted against a Master Coral database comprised of both genome-derived predicted gene models and transcriptomes spanning a wide diversity of coral families using BlastX v. 2.2.27[58,60−62]. Transcripts with less than 95% identity to this Master Coral database and shorter than 150 bp were filtered out. Next, the program TransDecoder v. 5.5.0 (https://github.com/TransDecoder/TransDecoder/wiki) was used to first extract the longest open reading frame (ORF) from each transcript and then to generate a predicted peptide sequence from this ORF, resulting in predicted proteomes for each of these three species. Sequences with high sequence similarity within each proteome were then collapsed using cd-hit v. 4.8.1[63] using default parameters. The resulting sequences were extracted from the initial assembly to generate coral-only reference transcriptomes. The completeness of the resulting assemblies was assessed with Benchmarking Universal Single Copy Orthologs (BUSCO) v. 5.2.2[64]. Assembly metrics can be found in Table 1. Finally, coral host and dominant symbiont transcripts were annotated with reviewed UniprotKB/Swiss-Prot Entry IDs using BlastX v. 2.2.27[65] using an evalue cutoff of 1.0e⁻⁶ (Supplementary Data 4).

### Isolation and quantification of coral and Symbiodiniaceae reads
BBsplit v. 38.90 was used to separate out coral, Symbiodiniaceae, and non-coral/non-Symbiodiniaceae reads using coral-only and Symbiodiniaceae reference transcriptomes under default parameters. Symbiodiniaceae transcriptomes representing the genera *Symbiodinium, Breviolum, Cladocopium*, and *Durusdinium* were sourced from previous publications (Table 1). The binning statistics output from BBSplit (Supplementary Information−Fig. 1, Supplementary Data 1) was used to assess which genera of Symbiodiniaceae was dominant within each sample, referred in the text as the "dominant symbiont." Coral and dominant symbiont reads were mapped to their respective transcriptome and quantified using the program Salmon v. 1.5.2[66,67] with default parameters used for corals and a kmer value of 23 for the dominant symbiont.

### Differential expression analysis of coral and dominant symbiont transcripts
Coral host and dominant symbiont transcript abundance was imported into R Studio v. 2022.02.2 and length-normalized using the package TXimport v. 1.16.1[68]. The remaining transcripts were regularized log (rlog) transformed and tested for differential expression using the package DESeq2 v. 1.30.1[69] with the design *-genotype + treatment* in the host and *-host_species + treatment* in the dominant symbiont (Supplementary Data 5 and 6). Expression profiles within a species or symbiont lineage were filtered for low abundance by removing transcripts with an average rlog expression <10. DEGs were identified in each species as those with a statistically significant difference in expression between control and SCTLD-treated corals (Wald test; padj ≤ 0.05). Principal component analysis (PCA) was performed to identify outliers in both the host and symbiont datasets and to illustrate the spatial relationships of gene expression across samples using the R package PCAtools v. 4.2.0[70] (Supplementary Information−Fig. 5). One sample within the host expression dataset (Pstr_d8) was identified as an outlier and removed from the study. One sample within the symbiont dataset (Oann_c2) was identified as an outlier and removed from the symbiont analysis.

### Gene Ontology enrichment analysis
Gene Ontology (GO) enrichment analyses were conducted using adaptive clustering of GO categories and Mann-Whitney U tests based on log2FoldChange values (GO_MWU, https://github.com/z0on/GO_MWU)[71]. The function "reduce_overlap" from the R package GOplot v. 1.0.2[71] was used to find the top 5 enriched non-redundant Biological Process (BP) and Molecular Function (MF) terms within each species (Supplementary Information−Fig. 3)[72].

To enumerate the number of DEGs in each species involved in immunity, the number of DEGs with GO terms containing at least one of the following words were counted: "immune", "immunity", "virus", and "viral".

### Differential expression analysis of relevant coral homologs
To compare rlog transformed expression of inferred homologs (transcripts with the same annotation) across coral species, species length-normalized counts matrices from Tximport were first merged by Uniprot Entry ID across all five coral species. Rlog normalized expression of all species' homologs was obtained by running this five-species count matrix through DESeq2 using the design *-species + treatment*, removing homologs with average rlog expression <10. Homologs involved in immunity and/or viral infection were isolated by pulling out those with GO terms containing the words "immune", "immunity", "virus" and "viral". Homologs involved in extracellular matrix organization were isolated by pulling out those with GO terms containing the words "extracellular matrix". A one-way ANOVA followed by TukeyHSD tests were run on these homologs to identify those with significant differential expression between disease states

(Supplementary Data 7). Each significant homolog's protein domains were identified and compared across species to confirm homology by uploading the protein-coding sequence associated with each transcript to Interpro (https://www.ebi.ac.uk/interpro/) (Supplementary Data 8).

## Differential expression analysis of relevant coral orthologs

To compare rlog transformed expression of orthologs across coral species, Orthofinder was used to identify groups of orthologous genes across the predicted proteomes of all five coral species, referred to as "orthogroups"[25,73]. Orthogroup count matrices were generated for each species using Tximport, and then were merged by orthogroup ID across all five coral species. Rlog normalized expression of all species' orthogroups was obtained by running this five-species count matrix through DESeq2 using the design -*species + treatment*, removing orthogroups with average rlog expression <10. Each orthogroup was assigned a function using the first *M. cavernosa* transcript in the orthogroup and matching it to the *M. cavernosa* transcriptome annotation. These annotated orthogroups are referred to in the text as "orthologs". Orthologs involved in immunity were isolated by pulling out those with GO terms containing the words "immune", "immunity", "virus" and "viral". Orthologs involved in extracellular matrix organization were isolated by pulling out those with GO terms containing the word "extracellular matrix". A one-way ANOVA followed by TukeyHSD tests were run on these orthologs to identify those with significant differential expression between disease states (Supplementary Data 9).

## Coral *rab7* correlations to symbiodiniaceae genes

To further investigate the biological processes of the intracellular symbionts that may be triggering symbiophagy, the expression of coral *rab7* was correlated to the expression of three photosystem genes identified as DEGs in *Breviolum* and *Durusdinium* (*psaA, psbB*, and *psbC*). We also correlated the expression of coral *rab7* to four other relevant genes identified from literature searches: two genes known to play a role in stress response (Superoxide dismutase and Heat shock protein 70), one gene involved in maintaining symbiosis with the coral host (Carbonic anhydrase), and a known apoptosis inducer (Apoptosis inducing factor 3)[48,74,75]. Expression correlations were separated by dominant symbiont genera within the sample and plotted in a correlation matrix (Fig. 7, Supplementary Data 14).

## Expression variance and evolution model using single-copy orthologs

The expression variance and evolution model was used to identify lineage-specific and highly variable single-copy orthologs across coral species and symbiont genera. Orthofinder was used to identify single-copy orthologs across the five coral species and across the four symbiont genera. The single-copy orthologs were annotated with BlastP v. 2.2.27[65] using the protein sequences from a representative species (*M. cavernosa* for the coral orthologs and *D. trenchii* for the symbiont orthologs). Coral host and dominant symbiont transcript abundance was Imported into R studio and length-normalized using Tximport using single-copy orthogroup IDs as the gene identifier. DESeq2 was run on both sets of single-copy ortholog counts to obtain rlog transformed expression values using the design -*species + treatment* in the coral dataset and -*genera + treatment* in the symbiont dataset. Single-copy orthologs with average rlog expression <10 were removed.

The rlog transformed expression of coral and symbiont single-copy orthologs was used in the EVE model[27] to test gene expression variation both among and within species (Supplementary Data 10 and 12). Genes exhibiting significant ($p \le 0.05$) expression plasticity or lineage-specific expression were identified. A one-way ANOVA followed by TukeyHSD tests were used to find significant differences in expression within the plastic orthologs across disease states (control

vs. exposed, exposed vs. infected, and control vs. infected) (Supplementary Data 11 and 13). Species average expression of lineage-specific genes was correlated to both species average lesion growth rate and species median relative risk of infection to identify single-copy orthologs contributing to adaptive differences in SCTLD susceptibility.

## Histology

Separate tissue samples were taken from each fragment for histology analyses and fixed in zinc-buffered formalin for 24 h, washed with fresh water for 24 h, and stored in 70% ethanol for transport to Louisiana State University, Baton Rouge, LA, USA. Samples were decalcified with a 1% HCl EDTA solution and stored in 70% ethanol until processed. Tissues were processed using a Lecia ASP6025, embedded in paraffin wax blocks on a Leica EG1150H embedding machine, and sectioned at five mm thickness on a Leica RM2125RTS microtome. Five serial sections were made 500 mm apart for each sample. Histological slides were stained with hematoxylin and eosin stain on a Leica ST5020, viewed on an Olympus BX41 microscope with an Olympus SC180 camera attachment.

Ten photomicrographs were taken at ×40 magnification (60,000 mm$^2$ focal area per image) per coral individual across all five serial sections. Photos were analyzed using ImageJ[76] software by overlaying a 12-cell grid on the image; each cell had an area of 5000 mm$^2$. A random number generator selected one grid-cell per photomicrograph. Within the grid-cell of interest, the areas of all symbiosomes and symbiont cells within symbiosomes were measured. Twenty-five symbiont and symbiosomes were measured for each coral colony. The condition of symbionts was noted.

## Amplicon sequencing of the internal transcribed Spacer 2 (ITS-2) region of Symbiodiniaceae rDNA

DNA was extracted using the ZymoBIOMICs DNA/RNA Miniprep kit (Zymo Research, CA), and the internal transcribed spacer-2 (ITS-2) region of Symbiodiniaceae rDNA was sequenced following Howe-Kerr et al.[77] using the primers SYM_VAR_5.8SII (5′ GAATTGCA-GAACTCCGTGAACC 3′) and SYM_VAR_REV (5′ CGGGTTCWCTTGTYT-GACTTCATGC 3′)[78] and sequenced on the Illumina MiSeq platform at the at the Oregon State University Center for Qualitative Life Sciences (CQLS, Corvallis, OR) following details outlined in Howe-Kerr et al.[77]. The resulting sequencing data were processed using Symportal[79].

## Reporting summary

Further information on research design is available in the Nature Portfolio Reporting Summary linked to this article.

# Data availability

The raw RNAseq data generated in this study have been deposited in the NCBI database under accession code PRJNA860922. The publicly available data used in this study include the transcriptomes for *Symbiodinium* CassKB8 (transcriptome assembly: http://medinalab.org/zoox/, accession number PRJNA80085), *Breviolum minutum* (transcriptome assembly: http://zoox.reefgenomics.org/download/, accession number PRJNA274852), *Cladocopium goreaui* (transcriptome assembly: http://ssid.reefgenomics.org/download/, accession number PRJNA307543) and *Durusdinium trenchii* (transcriptome assembly: https://datadryad.org/stash/dataset/doi:10.5061/dryad.12j173m, accession number PRJNA508937), as well as the genomes for *M. cavernosa* (genome assembly: https://matzlab.weebly.com/data-code.html, accession number PRJNA679067) and *O. faveolata* (genome assembly: https://www.ncbi.nlm.nih.gov/genome/13173?genome_assembly_id=311351, accession number PRJNA381078). The Master Coral database used in this study is available in a public Zenodo repository https://doi.org/10.5281/zenodo.7838980[80]. Supplementary data are provided as a Supplementary Data file. Source data are provided as a Source Data file. Source data are provided with this paper.

## Code availability

All shell scripts and R code used in this study are available in a Github repository https://doi.org/10.5281/zenodo.7839042[81].

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

## Acknowledgements

The authors thank the facilities and diving support staff at the University of the Virgin Islands Center for Marine and Environmental Studies (CMES) and Adam Glahn, Daniel Mele, Danielle Lasseigne, Kathryn Cobleigh, Alex Gutting, Amanda Long, and Bradley Arrington for their assistance with field collections and conducting experiments. Samples were collected under permit #DFW19057U authorized by the Department of Planning and Natural Resources Coastal Zone Management. This is CMES contribution #256. This work was funded by a National Science Foundation (NSF) VI EPSCoR 0814417 and 1946412 and NSF (Biological Oceanography) award numbers 1928753 to M.E.B. and T.B.S., 1928609 to A.M.S.C., 1928817 to E.M.M., 1928771 to L.D.M., 1927277 to D.M.H. as well as 1928761 and 1938112 to A.A., NSF EEID award number 2109622 to M.E.B., A.A., L.D.M., D.M.H., and A.M.S.C., and a NOAA OAR Cooperative Institutes award to A.A. (#NA19OAR4320074).

## Author contributions

Conceptualization: T.B.S., A.A., E.M.M., D.M.H., A.M.S.C., M.E.B., L.D.M. Sample collection: T.B.S., M.E.B., S.S.M., A.M.S.C. Experiment execution: A.J.V., N.J.M., B.A.D., S.S.M., A.M.S.C., M.E.B. RNA extraction and processing: K.M.B., with input from N.J.M. and B.A.D. Data analysis: K.M.B. and E.V.B., with input from M.E., N.J.M., and B.A.D. ITS2 sequencing and analysis: A.J.V., C.E.K., and A.M.S.C. Histology: A.R. and D.M.H. Manuscript writing: K.M.B. and L.D.M., with editing contributions from all other authors.

## Competing interests

The authors declare no competing interests.
