## [Peer Review File · Nature Communications]

Stony Coral Tissue Loss Disease Induces Transcriptional Signatures of in situ Degradation of Dysfunctional SymbiodiniaceaeREVIEWER COMMENTS

Reviewer #1 (Remarks to the Author):

Beavers et al. have described an excellent study on the transcriptional signatures of the devastating coral disease SCTLD. They have identified a variety of DEGs from transmission experiment corals as well as within their associated endosymbionts. There was a lot to unpack here but I found it to be a well-written paper, which made it much easier to digest. Most of my comments on concepts that could be expanded upon in the discussion of their various (and exciting) results. I didn't have any issues with the methods or data analysis, in fact, I found them to be much cleaner and controlled compared to other SCTLD studies I've seen.

- Some general points that can be expanded upon in the discussion:

(1) more comparison to the transcriptomics study done in FL; specifically, if the signatures suggest it may be the same disease (FL vs USVI).

(2) while I agree the results presented here does suggest the involvement of a viral pathogen, I still think it should be made clear that there is still a possibility of other microbes being at play. Specifically, a primary viral infection and bacterial secondary infection. This is hinted at, but I think all the potential possibilities should be laid out more clearly. Also, this may explain why the antibiotic treatments tend to work. However, this should be kept minimal, but at least mentioned.

(3) greater emphasis on the relationship between endosymbiont genera and their host susceptibility to SCTLD. Are there patterns? Links to DEGs and disease ecology?

Specific points

-Line 288: These viruses seem to be associated with both diseased and healthy corals. I suggest revision of this statement.

-Line 312: An important note is that this species was initially found to be resistant to SCTLD in FL years ago in Aeby et al. 2019.

-All figures: If possible can alternative color pallets be used that don't make this difficult for those that have color-blindness? At least not making the main colors red and green (e.g., figure 2). That would be much appreciated.

-Figure 1: some of the text are incredibly small. Especially some of the numbers. Can these be enlarged?

Reviewer #3 (Remarks to the Author):

Review

In the article entitled "Stony Coral Tissue Loss Disease Induces Transcriptional Signatures of in situ Degradation of Dysfunctional Symbiodiniaceae" Beavers et al. present the results of an integrative study aiming at understanding the transcriptional mechanisms associated to coral susceptibility/tolerance/resistance to SCTLD. To address this question authors performed Dual RNA-seq (cnidarian host and Symbiodiniaceae symbiont) on 5 coral holobiont representing 5 cnidarian and 4 Symbiodiniaceae species exposed or not and infected or not by SCTLD. These transcriptomic data in association to histology were analyzed with appropriate bioinformatics and statistical methods. The results obtained are very interesting, well discuss and will interest a large community.

My recommendation is that this article deserves a publication in Nature Communication but some modifications, clarifications and addition must be done before. Please see my major and minor comments for details.

Major comment

1 The infection experiment is in appearance quite simple and balanced but the results obtain are very very complex since many variable and covariate can combine. For example, for one coral species you can ended with, 8 different genotype, 2 different Symbiodiniaceae species, and 3 different "treatment" (control, diseases exposed and diseases infected). This complexity is not well describe in the MS (there is no result section dedicated to the results of the experimental infection)

and it was very challenging for me to really understand, what was compared to what, if the comparison were properly balanced in terms of symbiont species, host genotype etc. I believe that that a dedicated result section for the experimental infection is essential to enable a perfect understanding of the results subsequently shown. This section need to also clarify what samples were selected for sequencing. Indeed based on the m&ms we expect to see sequenced 80 libraries while there are "only" 52 samples that were finally sequenced, who and why stayed a mystery for me while I was reviewing the paper.

2 For the GO term enrichment analysis it is not clear for me how ShinyGo was used for the data set presented here. When I read the paper I understand that the ref list used as control list (named backroung in shyniGO documentation) to see if there is enrichment was the one for human. This ref list provides the numbers of genes per GO category and it is that numbers that are used for the statistical test. Since the number of genes, there putative function, the differences in terms of multigenic families etc etc are highly different between human and corals I think that shyniGO human ref-list is not. This point is well highlighted in shinyGO documentation: "We highly recommend users upload a list of genes as the background. These could be all the genes passed a low filter in RNA-seq. If background genes are not uploaded, the default is to use all protein-coding genes. Alternatively, you can check the box next to 'Use pathway database for gene counts', which will calculate background genes as the total unique number of genes in pathway database that users choose. As some pathway database can be huge and have genes not properly converted, we limit the total number to between 5000 and 30,000. When this option is used, any genes in user's original gene list but not in the pathway database will also be ignored." Author should redo this analysis using with a dedicated ref_list or a tool enabling to create the ref and the test list from their own species if ShinyGO can't. Alternatively it would be very interesting and more powerful to use tools enabling to weight differently each gene; GO_MWU for example is an interesting tool for that.

3 Line 544-546 I had some difficulties to understand what authors want to do here. Identifying genes present in all species and individuals? If yes, it is more stringent to use the results obtained with OrthoFinder rather that the Uniprot ID obtain from a blast. Two different Query can ended whit significant blast results pointing toward the same Uniprot ID however it do not mean that this two query are the "same gene" since the similarity can come from different part of the reference sequence. If you want to keep this method please precise in the results when you talk about these genes that there are genes sharing the same annotation to avoid confusion with orthologous genes.

Minor comment

- Line 95 a result section dedicated to the description of the experimental infection is necessary and will help the understanding of the MS
- Line 97 to 99 it would be nice to provide in a table or a figure summarizing the experimental design. It should show the number of samples that were finally sequenced for each species, genet, control, diseases exposed and diseases infected treatments.
- Line 458-460 precise how many genet of Diploria were used as diseases donor or control donor
- Line 485-504 the origin and bioinformatics treatments performed on each transcriptome should be better described; it is difficult to understand what was done on/from each transcriptome. For example, the Transcriptome of M cavernosa is not mentioned in this section.
- Line 518-519 I propose to move the annotation information to the transcriptome assembly section and to rename it Coral transcriptome assembly and annotation
- Line 526 a log2FC threshold value seems to be missing at the end of the sentence "Significantly differentially expressed genes were identified in each species as those with a padj \leq 0.05 for log2 fold change.
- Line 527 to 529 I recommend to move this part describing a functional analysis of the dataset rather than a differential gene expression analysis to the section GO ontology enrichment analysis.
- Line 546 DESeq2 and not DESeq I guess
- Line 557-560 precise in which genera these genes were DEG as you did it for the photosystem encoding protein gene in Breviolium and Durusdinium
- Line 618 to 621 all de novo transcriptome assembly and annotation performed in this study should be provided as supplementary material or deposited in a public library.
- Line 262-266, since the species differences in terms of "gene constitutive expression"

(comparison between species in the control treatment) it was not possible to understand this conclusion

- Line 264-266, same comment: since the result of the DE analysis was not presented for all the comparison performed it is not possible to understand this conclusion. Authors should consider a way to show these DE results for all the comparison performed.
- Line 305 I suggest to precise that the opportunistic infection is a secondary infection by opportunistic bacteria
- Line 317-320 a little bit more explanation about the link between ECM activation, apoptosis and necrotic tissue degradation would help the reader to better understand.
- Line 355-359, again a problem due to the absence of results linked to basal expression analysis
- -line 385-386 I don't understand how authors can make a link between overexcitation of photosynthetic apparatus and a worse outcome for SCTL. This hypothesis should be more explain or removed.
- Line 414-417 this conclusion/hypothesis is not very clear for me. The initial question addressed in this paragraph is why symbiophagy rather than exocytosis. If I well understand the author propose that the coral host use symbiophagy rather than exocytosis to remove infected symbiont in order to also use the symbiont as a source of organic matter. An alternative/complementary hypothesis would be that autophagy is also induced to mount the antiviral response. Indeed autophagy is a well-known mechanism involved in antiviral response and this proposition is strengthen by the fact that the coral mount an antiviral immune response which means that he sense the viral infection even if it infect the zoo. For me the symbiophagy would be more probably a defense response mechanism rather than a starvation mechanism, no?
- 435-442 this paragraph is for me out of scope.

Figure and tables

- Table 1: for *O. annularis* mention genome guided instead of de novo. For *cavernosa*, *natans* and *astreoides* I think that mentioning how these transcriptome were created de novo or genome guided would be more informative that the ref, you can cite the ref in the legend. Same comment for the *symbiodiniacea*
- The table 1 should provide some basic metrics about the annotations of each transcriptome, such as the % of sequence presenting a significant similarity with a protein of known function, the % of sequence presenting a significant similarity with a protein of unknown function, the % of sequence presenting no significant similarity.
- Fig 1F , the Venn Diagram shows that 26 genes were significantly differentially expressed between diseases states while in the results section it is mentioned that there are 51 (26 are the one with a significant annotation if I follow well).
- Supplementary files showing all the DEseq2 results are needed
- Supplementary files showing the EVE results are needed
- Supplementary files showing PCA obtained from the DE analysis are needed
- Supplementary files with the annotation of all transcriptomes used are needed

Congratulation for this super interesting work
Jeremie Vidal-Dupiol

Reviewer 1

Beavers et al. have described an excellent study on the transcriptional signatures of the devastating coral disease SCTLD. They have identified a variety of DEGs from transmission experiment corals as well as within their associated endosymbionts. There was a lot to unpack here but I found it to be a well-written paper, which made it much easier to digest. Most of my comments on concepts that could be expanded upon in the discussion of their various (and exciting) results. I didn't have any issues with the methods or data analysis, in fact, I found them to be much cleaner and controlled compared to other SCTLD studies I've seen.

We thank the reviewer for these supportive comments. We are very pleased our manuscript received this response and look forward to being able to share it with the larger scientific community.

- Some general points that can be expanded upon in the discussion:

1. more comparison to the transcriptomics study done in FL; specifically, if the signatures suggest is may be the same disease (FL vs USVI).

Thank you for this comment. Where appropriate we did bolster the comparison to the FL study, especially when discussing specific gene comparisons that indeed point to a similar disease. Since the two papers differ in their analyses and scope, making a specific link between FL and UVI diseases may require a meta-analysis which we feel is beyond the scope of this paper. However, this is something we are working on in the lab.

2. while I agree the results presented here does suggest the involvement of a viral pathogen, I still think it should be made clear that there is still a possibility of other microbes being at play. Specifically, a primary viral infection and bacterial secondary infection. This is hinted at, but I think all the potential possibilities should be laid out more clearly. Also, this may explain why the antibiotic treatments tend to work. However, this should be kept minimal, but at least mentioned.

Thank you for this comment. The discussion has been expanded to address this point at lines 383-435.

“Downregulation of *Dmbt1* in SCTLD-infected corals may therefore render corals unable to maintain mucosal microbial homeostasis, leading to a loss of the protective capabilities of their surface mucus layer and making them susceptible to secondary infection by opportunistic bacteria. These results explain why antibiotic treatment is sufficient to arrest treated SCTLD lesions, but ineffective at preventing new lesion appearance on other parts of the same coral colony³¹.”

3. greater emphasis on the relationship between endosymbiont genera and their host susceptibility to SCTLD. Are there patterns? Links to DEGs and disease ecology?

Thank you for this comment, we realized that our paragraph addressing this issue could be made clearer. We re-wrote paragraph starting on line 525 to make these connections stronger.

“Because *in situ* degradation of endosymbionts represents a host innate immune response to compromised symbionts⁴⁵, we investigated transcriptional evidence of symbiont impairment. Within the symbiont DEGs, we see evidence of photosynthesis dysfunction in both the downregulation of photosystem genes in *Breviolum* spp., as well as the upregulation of those same genes in *Durusdinium* spp. In fact, all the genes upregulated by *Durusdinium* symbionts are involved in energy production, likely indicative of photosystem overexcitation. Furthermore, the samples dominated by *Durusdinium* spp. displayed the fastest lesion growth rates in our study, indicating that symbiont photosystem overexcitation may lead to worse outcomes for SCTLD infected corals.”

Specific points

1. Line 288: These viruses seem to be associated with both diseased and healthy corals. I suggest revision of this statement.

Thank you for this comment. During revisions, this paragraph was removed and replaced with a discussion regarding genes from the new ortholog analysis.

2. Line 312: An important note is that this species was initially found to be resistant to SCTLD in FL years ago in Aeby et al. 2019.

We apologize for the confusion. This sentence we rewritten and the statement about *Porites astreoides* was removed.

3. All figures: If possible can alternative color pallets be used that don't make this difficult for those that have color-blindness? At least not making the main colors red and green (e.g., figure 2). That would be much appreciated.

We have modified the figures to be more colorblind-accessible by replacing reds with orange when green is present in the same figure.

4. Figure 1: some of the text are incredibly small. Especially some of the numbers. Can these be enlarged?

All text size has been enlarged in Figure 1

Reviewer 3

In the article entitled “Stony Coral Tissue Loss Disease Induces Transcriptional Signatures of in situ Degradation of Dysfunctional Symbiodiniaceae” Beavers et al. present the results of an integrative study aiming at understanding the transcriptional mechanisms associated to coral susceptibility/tolerance/resistance to SCTLD. To address this question authors performed Dual RNA-seq (cnidarian host and Symbiodiniaceae symbiont) on 5 coral holobiont representing 5 cnidarian and 4 Symbiodiniaceae species exposed or not and infected or not by SCTLD. These transcriptomic data in association to histology were analyzed with appropriate bioinformatics and statistical methods. The results obtained are very interesting, well discuss and will interest a large community. My recommendation is that this article deserves a publication in Nature Communication but some modifications, clarifications and addition must be done before. Please see my major and minor comments for details.

We thank reviewer 3 for these comments and their support of our paper. We were able to address all the reviewer comments which strengthened our analysis and interpretation.

Major Comments

Results

1. The infection experiment is in appearance quite simple and balanced but the results obtain are very very complex since many variable and covariate can combine. For example, for one coral species you can ended with, 8 different genotype, 2 different Symbiodiniaceae species, and 3 different “treatment” (control, diseases exposed and diseases infected). This complexity is not well describe in the MS (there is no result section dedicated to the results of the experimental infection) and it was very challenging for me to really understand, what was compared to what, if the comparison were properly balanced in terms of symbiont species, host genotype etc. I believe that that a dedicated result section for the experimental infection is essential to enable a perfect understanding of the results subsequently shown. This section need to also clarify what samples were selected for sequencing. Indeed based on the m&ms we expect to see sequenced 80 libraries while there are “only” 52 samples that were finally sequenced, who and why stayed a mystery for me while I was reviewing the paper. We do apologize for any difficulty the readers may have had understanding the results of the transmission study. As such, we added the results of the experiment to the beginning of the results section (lines 101-114). We also created a metadata file with information on every sample, including information on which samples were sequenced and which samples were removed as outliers and why. We have also added a sentence to the methods that explains why only 52 samples were sequenced at lines 668-669: “The 52 samples that passed quality assessment (RIN numbers ≥ 7) were sent to Novogene Co., LTD (Beijing, China).” We feel that this, in addition to the results of the transmission summarized in Fig 1A, would help the reader understand the context better.

2. For the GO term enrichment analysis it is not clear for me how ShinyGo was used for the data set presented here. When I read the paper I understand that the ref list used as control list (named background in shinyGO documentation) to see if there is enrichment was the one for human. This ref list provides the numbers of genes per GO category and it is that numbers that are used for the statistical test. Since the number of genes, there putative function, the differences in terms of multigenic families etc etc are highly different between human and corals I think that shinyGO human ref-list is not. This point is well highlighted in shinyGO documentation: “We highly recommend users upload a list of genes as the background. These could be all the genes passed a low filter in RNA-seq. If background genes are not uploaded, the default is to use all protein-coding genes. Alternatively, you can check the box next to 'Use pathway database for gene counts', which will calculate background genes as the total unique number of genes in pathway database that users choose. As some pathway database can be huge and have genes not properly converted, we limit the total number to between 5000 and 30,000. When this option is used, any genes in user's original gene list but not in the pathway database will also be ignored.” Author should redo this analysis using with a dedicated ref_list or a tool enabling to create the ref and the test list from their own species if ShinyGO can't. Alternatively it would be very interesting and more powerful to use tools enabling to weight differently each gene; GO_MWU for example is an interesting tool for that.

Upon further reflection, we did concur with the reviewer that using ShinyGO has limitations in that the functional enrichment is based on human function, even when a background set of genes is provided. To overcome this limitation, we took this recommendation and redid the GO enrichment analysis using the suggested tool, GO_MWU. With this new analysis, we no longer find “extracellular matrix structural constituent” as a commonly enriched GO term across the 5 coral species. We did find enrichment of “DNA metabolic process” in both highly susceptible species, *C. natans* and *O. annularis*. We also see enrichment of “peptide biosynthetic process” and “cell adhesion” in both moderately susceptible species, *P. strigosa* and *P. astreoides*. *M. cavernosa*, the species with the lowest susceptibility to SCTLD, has unique GO enrichments, such as “protein folding” and “transporter activity.” We have updated the methods, supplementary files, and the manuscript to reflect this change in analysis.

3. Line 544-546 I had some difficulties to understand what authors want to do here. Identifying genes present in all species and individuals? If yes, it is more stringent to use the results obtained with OrthoFinder rather than the Uniprot ID obtain from a blast. Two different Query can ended with significant blast results pointing toward the same Uniprot ID however it do not mean that this two query are the “same gene” since the similarity can come from different part of the reference sequence. If you want to keep this method please be precise in the results when you talk about these genes that there are genes sharing the same annotation to avoid confusion with orthologous genes. Thank you for this comment and we understand the concerns. Our previous work (Dimos et al. 2022) showed that immune genes in corals belong to rapidly expanding gene families and therefore likely do not exist as single-copy orthologs. We therefore

used this BLAST homology-based approach to make sure we did not exclude important immune genes from our analysis. To confirm the identity of these genes, we conducted a protein domain analysis using InterPro. Consequently, we removed genes from the manuscript that do not have same domain structure across all 5 coral species. In the text, we refer to those genes that passed our stringent domain homology analysis as “homologs”. We have added the protein domain analysis of the homologs to the supplementary material.

However, we do agree with the reviewer that using BLAST annotations to identify genes with the same function does have its caveats. To this extent, we performed an additional immune analysis using orthogroups that contain single- AND multi-copy orthologous genes, referred to in the text as “orthologs”. We have replaced figure 2 to represent the results from this ortholog analysis. The conclusions remained the same, but we agree the readers will have more confidence in this analysis.

To avoid any confusion about what analysis we are showing we have updated the methods, the results, and the figure legends to explicitly state whether homologs or orthologs are being talked about. Single-copy orthologs were still used for the EVE analysis, which didn’t change.

Minor Comments

1. Line 95 a result section dedicated to the description of the experimental infection is necessary and will help the understanding of the MS
We have added a paragraph to the results section summarizing the experimental results from Meiling et al. 2021 (lines 101-114).
2. Line 97 to 99 it would be nice to provide in a table or a figure summarizing the experimental design. It should show the number of samples that were finally sequenced for each species, genet, control, diseases exposed and diseases infected treatments.
We have added more detail to our supplementary material and have increased the reference to Meiling et al. 2021, where all experimental results have been published.
3. Line 262-266, since the species differences in terms of “gene constitutive expression” (comparison between species in the control treatment) it was not possible to understand this conclusion
We apologize for this confusing wording; this is indeed an output of the EVE model. The input of EVE is the rlog normalized expression of all single-copy orthologs from all samples and treatments, including controls, exposed and infected samples. The output of EVE includes a beta value for each gene as well as a “shared beta” value, which represents the threshold of stabilizing or no selection acting on expression level. Genes with beta values above the “shared beta” represent genes with high levels of expression plasticity, while genes with beta values below the “shared beta” represent genes with low expression plasticity within a species (but high variation between species). The

latter set of genes are what we refer to in the text as “lineage-specific”, and the expression of these genes does not change between disease states. We have shown that the species-average expression of these lineage-specific genes are strongly correlated to species average lesion growth rate as well as species median relative risk. We have added a sentence at line 845 to help clarify this: “The rlog transformed expression of coral and symbiont single-copy orthologs was used in the EVE model²⁶ to test gene expression variation both among and within species.” We have also removed the word “constitutive” from the manuscript to avoid confusion.

4. Line 264-266, same comment: since the result of the DE analysis was not presented for all the comparison performed it is not possible to understand this conclusion. Authors should consider a way to show these DE results for all the comparison performed. We apologize for the confusion. We have adjusted this paragraph to clarify where each result came from beginning at line 352: “First, we found that SCTL D infection induced expression of homologous and orthologous genes involved in immunity, apoptosis, and ECM structure in all five coral species. Second, we showed that species-level SCTL D susceptibility was correlated to lineage-specific differences in expression of single-copy orthologs involved in vesicular trafficking and signal transduction. Third, we found evidence that SCTL D exposure (without visible lesions) induces major expression shifts in the highly variable Symbiodiniaceae single-copy orthologs, but not in those of the coral animal.”
5. Line 305 I suggest to precise that the opportunistic infection is a secondary infection by opportunistic bacteria
We corrected this and this comment is related to a comment from Reviewer 1.
6. Line 317-320 a little bit more explanation about the link between ECM activation, apoptosis and necrotic tissue degradation would help the reader to better understand. We have expanded this section for clarity beginning at line 437: “In addition, our results indicate a significant upregulation of apoptosis and related stress-response processes in SCTL D-infected corals relative to controls. *SMAD6*, a member of the transforming growth factor (TGF- β) signaling pathway, exhibited a strong decrease in expression between control and infected corals. Knockdown of this protein has been shown to increase apoptosis and inhibit cell cycle progression in human cells³². Similarly, *ALOX5*, a proinflammatory gene hypothesized to play a role in the phagocytosis of apoptotic bodies in regenerating Hydra³² showed a concomitant increase in expression in infected corals. We also found significant upregulation of the antioxidant *sodA*, an established coral immune response antioxidant also involved in symbiosis breakdown^{34,35}. Finally, we observed strong downregulation of multiple collagen genes in infected corals, suggesting a stress-induced decrease in ECM structural proteins not immediately needed for cell survival.”
7. Line 355-359, again a problem due to the absence of results linked to basal expression analysis

See above comment (3) for line 262-266.

8. line 385-386 I don't understand how authors can make a link between overexcitation of photosynthetic apparatus and a worse outcome for SCTL. This hypothesis should be more explain or removed.

We have rephrased this paragraph for clarity beginning at line 532: "In fact, all the genes upregulated by *Durusdinium* symbionts are involved in energy production, likely indicative of photosystem overexcitation. The samples dominant in *Durusdinium* spp. displayed the fastest lesion growth rates in our study, indicating that symbiont photosystem overexcitation may lead to worse outcomes for SCTL infected corals."

9. Line 414-417 this conclusion/hypothesis is not very clear for me. The initial question addressed in this paragraph is why symbiophagy rather than exocytosis. If I well understand the author propose that the coral host use symbiophagy rather than exocytosis to remove infected symbiont in order to also use the symbiont as a source of organic matter. An alternative/complementary hypothesis would be that autophagy is also induced to mount the antiviral response. Indeed autophagy is a well-known mechanism involved in antiviral response and this proposition is strengthen by the fact that the coral mount an antiviral immune response which means that he sense the viral infection even if it infect the zoo. For me the symbiophagy would be more probably a defense response mechanism rather than a starvation mechanism, no?

We apologize if there was any confusion. Our intent with pointing out genes indicative of starvation was meant as further evidence that the symbiont is not doing its job, likely due to viral infection, that does trigger the host immune response against the symbiont. It was meant as complimentary to immune response and not an either/or scenario. We have expanded on this beginning on line 591:

"Because corals derive up to 100% of their daily metabolic requirements from their Symbiodiniaceae⁵¹, it is likely that the concomitant expression decrease in *Tpm4* and increase in *Btg1* are indicative of coral starvation due to inadequate photosynthate transfer from symbionts. Taken together with the increased expression of coral immune- and stress-response genes, this provides evidence that SCTL-infected corals are mounting an autophagic immune response against dysfunctional symbionts and/or their pathogens."

10. 435-442 this paragraph is for me out of scope.

Upon further reflection, we agree and have removed the paragraph.

11. Line 458-460 precise how many genet of *Diploria* were used as diseases donor or control donor

This has been added to the Methods beginning on line 649: "Eight genets of healthy *D. labyrinthiformis* were used as control donor corals and eight genets of diseased *D. labyrinthiformis* were used as SCTL donor corals."

12. Line 485-504 the origin and bioinformatics treatments performed on each transcriptome should be better described; it is difficult to understand what was done on/from each transcriptome. For example, the Transcriptome of *M cavernosa* is not mentioned in this section.

Yes, we have fixed this issue and it is addressed in table 1.

13. Line 518-519 I propose to move the annotation information to the transcriptome assembly section and to rename it Coral transcriptome assembly and annotation
Good idea, it is done.

14. Line 526 a log₂FC threshold value seems to be missing at the end of the sentence “Significantly differentially expressed genes were identified in each species as those with a $p_{adj} \leq 0.05$ for log₂ fold change.

This sentence re-worded for clarity beginning on line 725: “DEGs were identified in each species as those with a statistically significant difference in expression between control and SCTLD-treated corals (Wald test; $p_{adj} \leq 0.05$).”

15. Line 527 to 529 I recommend to move this part describing a functional analysis of the dataset rather than a differential gene expression analysis to the section GO ontology enrichment analysis.

Done

16. Line 546 DESeq2 and not DESeq I guess

Yes, corrected.

17. Line 557-560 precise in which genera these genes were DEG as you did it for the photosystem encoding protein gene in *Breviolum* and *Durusdinium*

As mentioned above we have rewritten this part to improve the clarity beginning on line 793:

“To further investigate the biological processes of the intracellular symbionts that may be triggering symbiophagy, the expression of coral *rab7* was correlated to the expression of three photosystem genes identified as DEGs in *Breviolum* and *Durusdinium* (*psaA*, *psbB*, and *psbC*). We also correlated the expression of coral *rab7* to seven other relevant genes relevant to symbiosis and stress response identified from literature searches: two plasma membrane transporters (High affinity nitrate transporter 2.5 and Two pore calcium channel protein 1), two genes known to play a role in stress response (Superoxide dismutase and Heat shock protein 70), one gene involved in maintaining symbiosis with the coral host (Carbonic anhydrase), and a known apoptosis inducer (Apoptosis inducing factor 3)^{47,73–76}.”

18. Line 618 to 621 all de novo transcriptome assembly and annotation performed in this study should be provided as supplementary material or deposited in a public library. We have uploaded the raw sequences file from Novogene on to NCBI and have also uploaded a detailed Rmarkdown code describing the step-by-step assembly process on github (https://github.com/kbeavz/SCTLD-Transmission-Experiment-USVI/blob/main/Command_Line_Code.Rmd). We have also added each transcriptome's annotation to the supplementary material.

Figures and Tables

1. Fig 1F , the Venn Diagram shows that 26 genes were significantly differentially expressed between diseases states while in the results section it is mentioned that there are 51 (26 are the one with a significant annotation if I follow well). Both the coral and the symbiont venn diagrams have been changed to reflect the total number of single-copy orthologs differentially expressed between disease states, not just the annotated ones
2. Table 1: for *O. annularis* mention genome guided instead of de novo. For *cavernosa*, *natans* and *astreoides* I think that mentioning how these transcriptome were created de novo or genome guided would be more informative than the ref, you can cite the ref in the legend. Same comment for the *symbiodiniaceae*
These edits have been made to Table 1
3. The table 1 should provide some basic metrics about the annotations of each transcriptome, such as the % of sequence presenting a significant similarity with a protein of known function, the % of sequence presenting a significant similarity with a protein of unknown function, the % of sequence presenting no significant similarity. We have added a "Percent Annotated" column to table 1. As very few coral proteins are annotated with known function, we frequently infer function from sequence homology. For this reason, we have decided not to include columns describing the percentage of proteins annotated with known or unknown function, as it may be misleading. We believe that the annotation evalue cutoff of $1.0e-06$ is sufficient in identifying sequences with known (or inferred) function, and the remaining unannotated sequences represent protein-coding sequences of unknown function.
4. Supplementary files showing all the DEseq2 results are needed
This has been added to the Supplementary Material
5. Supplementary files showing the EVE results are needed
This has been added to the Supplementary Material
6. Supplementary files showing PCA obtained from the DE analysis are needed
This has been added to the Supplementary Figures

7. Supplementary files with the annotation of all transcriptomes used are needed
This has been added to the Supplementary Material

Other corrections we made:

- Typo: E-value cutoff of $10e^{-6}$ has been corrected to the one used in the R code: $1.0e^{-6}$
- Typo: *O.annularis*'s median relative risk value has been corrected from 12.19 to 12.09
- Typo: In Figure 8 Pstr_d7 was labeled "exposed" but should be labeled "infected"
- While making revisions, we corrected a small error in the EVE analysis code. I have revised the Rmarkdown analysis and made the following changes in the manuscript:
 - There are 84 Lineage-Specific coral orthologs (instead of 85)
 - Of the 84, 75 were annotated with sufficient evalue (instead of 77)
 - The same 3 orthologs were found to be correlated to relative risk and the same and the same 5 orthologs were found to be correlated to lesion growth rate, so no changes were made to Fig. 3 or to its interpretation in the manuscript
 - There are 217 Highly Variable coral orthologs (instead of 216)
 - Of the 217, 78 show significant differences in expression between disease states
 - Of those 78, 53 have sufficient evalue (instead of 55)
 - The same 31 genes with the most significant differential expression between disease states ($p \leq 0.01$) remained the same, so no changes were made to Figure 4 or to its interpretation in the manuscript
 - There are 292 Lineage-specific symbiont orthologs (no change)
 - However, 4 of these were found to be significantly correlated to lesion growth rate ($p \leq 0.01$). These results have been added to the results section.
 - There are 1,212 highly variable symbiont orthologs (instead of 1,209)
 - Of the 1,212, 48 show significant differences in expression between disease states (instead of 51)
 - Of those 48, 24 have sufficient evalue ($1.0e^{-6}$, not 0.01) – Figure 6 has been re-made to reflect these changes

REVIEWERS' COMMENTS

Reviewer #1 (Remarks to the Author):

I have read over the responses to my comments and the revised paper, and I would like to thank the authors for their modifications. It does seem like some of my suggestions were beyond the scope of this study, and I would agree with that but I am excitedly awaiting the follow up! I don't have any further suggestions for this paper. I would be comfortable with publications of this revised manuscript.

Reviewer #3 (Remarks to the Author):

Dear authors,
congratulation for this very good work and thank you for having considering my comments. I think that this version of the MS is totally suitable for publication in Nature Communication
Best regards
Jeremie